# Crystal structures of the ATP-binding and ADP-release dwells of the V₁ rotary motor

Kano Suzuki[1], Kenji Mizutani[1,2,3], Shintaro Maruyama[1], Kazumi Shimono[4], Fabiana L. Imai[1], Eiro Muneyuki[5], Yoshimi Kakinuma[6], Yoshiko Ishizuka-Katsura[7], Mikako Shirouzu[7], Shigeyuki Yokoyama[8], Ichiro Yamato[3] & Takeshi Murata[1,2,7,9]

$V_1$-ATPases are highly conserved ATP-driven rotary molecular motors found in various membrane systems. We recently reported the crystal structures for the *Enterococcus hirae* $A_3B_3DF$ ($V_1$) complex, corresponding to the catalytic dwell state waiting for ATP hydrolysis. Here we present the crystal structures for two other dwell states obtained by soaking nucleotide-free $V_1$ crystals in ADP. In the presence of 20 μM ADP, two ADP molecules bind to two of three binding sites and cooperatively induce conformational changes of the third site to an ATP-binding mode, corresponding to the ATP-binding dwell. In the presence of 2 mM ADP, all nucleotide-binding sites are occupied by ADP to induce conformational changes corresponding to the ADP-release dwell. Based on these and previous findings, we propose a $V_1$-ATPase rotational mechanism model.

[1] Department of Chemistry, Graduate School of Science, Chiba University, 1-33 Yayoi-cho, Inage, Chiba 263-8522, Japan. [2] Molecular Chirality Research Center, Chiba University, 1-33 Yayoi-cho, Inage, Chiba 263-8522, Japan. [3] Department of Biological Science and Technology, Tokyo University of Science, 2641 Yamazaki, Noda-shi, Chiba 278-8510, Japan. [4] Faculty of Pharmaceutical Sciences, Toho University, 2-2-1 Miyama, Funabashi 274-8510, Japan. [5] Department of Physics, Faculty of Science and Engineering, Chuo University, 1-13-27 Kasuga, Tokyo 112-8551, Japan. [6] Laboratory of Molecular Physiology and Genetics, Faculty of Agriculture, Ehime University, 3-5-7 Tarumi, Matsuyama, Ehime 790-8566, Japan. [7] Division of Structural and Synthetic Biology, RIKEN Center for Life Science Technologies, 1-7-22 Suehiro-cho, Tsurumi, Yokohama 230-0045, Japan. [8] RIKEN Structural Biology Laboratory, 1-7-22 Suehiro-cho, Tsurumi, Yokohama 230-0045, Japan. [9] JST, PRESTO, 1-33 Yayoi-cho, Inage, Chiba 263-8522, Japan. Correspondence and requests for materials should be addressed to T.M. (email: t.murata@faculty.chiba-u.jp).

on-transporting rotary ATPases are divided into three types based on their function and taxonomic origin: F-, V- and A-type ATPases. F-ATPases function as ATP synthases in mitochondria, chloroplasts and oxidative bacteria[1]. V-ATPases function as proton pumps in acidic organelles and plasma membranes of eukaryotic cells[2]. A-ATPases function as ATP synthases similar to the F-ATPases in Archaea (the 'A' designation refers to Archaea), but the structure and subunit composition of A-ATPases are more similar to those of V-ATPases[3]. These ATPases possess similar overall structures consisting of a globular catalytic domain ($F_1$, $V_1$ or $A_1$) and a membrane-embedded ion-transporting domain ($F_o$, $V_o$ or $A_o$). These catalytic domains are similar rotary molecular motors, in which the central axis complexes rotate within pseudo-hexagonally arranged catalytic complexes powered by energy from ATP hydrolysis[1–6].

The rotational catalysis of $F_1$-ATPase has been investigated using structural analyses of bovine[7–11], yeast[12,13] and bacterial[14,15] samples, and by single-molecule dynamics studies of bacterial samples[16–21]. However, contradictory findings have been obtained depending on the methods, conditions and species, leading to controversy regarding the general rotational model of $F_1$ (refs 10,18). Recently, a model was proposed that could consistently explain both the structural and single-molecule data obtained for mammalian $F_1$-ATPase[22,23]. In this model, the central axis rotates at 120° per ATP molecule with three dwell states: waiting for ATP binding (ATP-binding dwell) at 0° (and 120°), waiting for $P_i$ release ($P_i$-release dwell) at 65°, and waiting for ATP hydrolysis (catalytic dwell) at 90° (see Fig. 1).

Similar $V_1$-ATPase experiments have been conducted using bacterial enzymes from *Thermus thermophilus*[24–28] and *E. hirae*[29–33]. These enzymes are sometimes called A-ATPases. However, they are derived from Eubacteria, rather than Archaea[3]. Furthermore, *E. hirae* V-ATPase physiologically functions as an ion pump, similar to eukaryotic V-ATPases[34–37], and is composed of nine subunits with amino acid sequences that are homologous to those of the corresponding subunits of eukaryotic V-ATPases[6,38]. Therefore, we believe the enzyme is a homologue of eukaryotic V-ATPases. We previously established the *in vitro* expression, purification and crystallization of *E. hirae* $V_1$-ATPase ($EhV_1$) from the $A_3B_3$ and DF complexes[29,30]. The crystal structures of the nucleotide-free and nucleotide-bound $A_3B_3$ ($eA_3B_3$ and $bA_3B_3$) and $V_1$ ($eV_1$ and $bV_1$) complexes revealed conformational changes of the $A_3B_3$ complex induced by the binding of nucleotides and the DF axis (Supplementary Fig. 1), suggesting that the $EhV_1$ structure corresponds to the catalytic dwell waiting for ATP hydrolysis in the rotary cycle[31]. We have also directly confirmed the unidirectional rotation of $EhV_1$ with single-molecule observations[32,33]. $EhV_1$ shows only three pausing positions separated by 120° at all ATP concentrations without distinct substeps, in contrast to that of $F_1$-ATPase[17,22]. This suggests that the ATP hydrolysis step(s), for example, ATP binding, phosphate bond cleavage, ADP release or $P_i$ release, is/are the rate-limiting step(s) in the three-pause rotation[39,40]. In this study, we performed experiments in which nucleotide-free $V_1$ crystals were soaked with AMP-PNP (non-hydrolysable ATP analog adenosine 5'-(β,γ-imino)triphosphate), ADP or phosphate, and obtained two previously unidentified crystal structures corresponding to the ATP-binding dwell and ADP-release dwell states in the rotary cycle of $EhV_1$. Our proposed rotational mechanism of $EhV_1$ based on these crystal structures is apparently different from those previously reported for $F_1$-ATPases[18–23] (see Fig. 1).

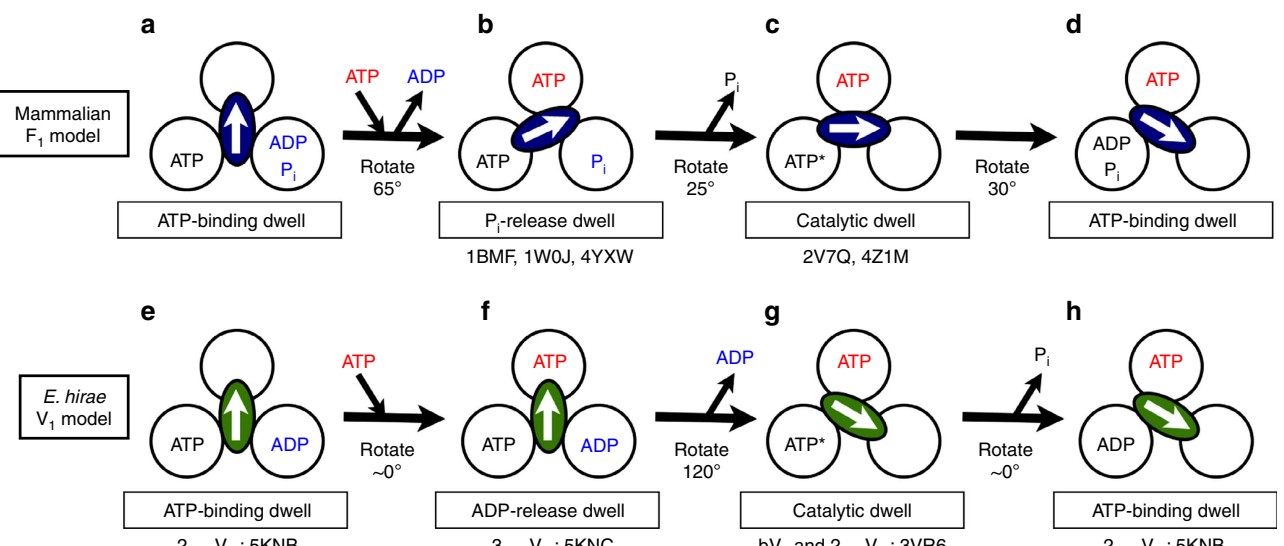

**Figure 1 | Coupling scheme for the 120° rotation and ATP hydrolysis of mammalian F₁- and *Enterococcus hirae* V₁-ATPases.** Each circle represents the chemical state of the nucleotide-binding site, viewed from the cytoplasmic side (that is, the N-terminal β-barrel side of $V_1$). The central arrows in the ellipses represent the orientation of the central axis beginning from the twelve o'clock position, which corresponds to the ATP-binding dwell (a waiting state for ATP binding). PDB ID numbers of the corresponding crystal structures are shown under the schemes. ATP* represents an ATP molecule that is committed to hydrolysis. (**a–d**) A model for mammalian $F_1$ (refs 22,23). ATP binding to the ATP-binding dwell (**a**) induces a 65° rotation concomitant with ADP release from another binding site and resulting conformational changes to the $P_i$-release dwell[7,9,23] (**b**). $P_i$ release induces a 25° rotation and consequent conformational changes to the catalytic dwell[11,23] (**c**), which is waiting for ATP hydrolysis. ATP* hydrolysis to produce ADP and $P_i$ induces a 30° rotation and conformational changes to the ATP-binding dwell (**d**). (**e–h**) A model for *E. hirae* $V_1$ (this study). ATP binding to the ATP-binding dwell (**e**) induces conformational changes to the ADP-release dwell (**f**) without an apparent rotational substep of the central axis. ADP release induces a 120° rotation and consequent conformational changes to the catalytic dwell (**g**). ATP* is hydrolysed to produce ADP and $P_i$, and the $P_i$ release induces conformational changes to the ATP-binding dwell state (**h**) without a rotational substep.

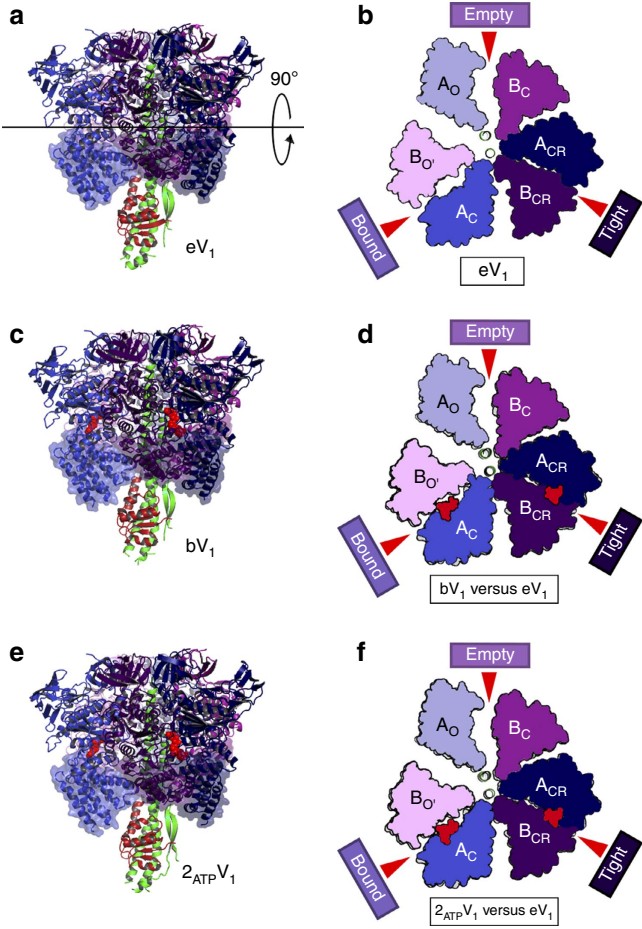

**Figure 2 | Structures of the nucleotide-free $V_1$ ($eV_1$) and 2AMP-PNP-bound $V_1$ complexes ($bV_1$ and $2_{ATP}V_1$).** (**a**) Side view of $eV_1$. (**b**) Top view of the C-terminal domain (transparent surface in **a**) from the cytoplasmic side. Open (O and O'; light), closed (C; dark) and closer (CR; darker) conformations of Eh-A and Eh-B are shown. Red arrows indicate the nucleotide-binding sites. (**c,e**) Side views of $bV_1$ (**c**) and $2_{ATP}V_1$ (**e**). (**d,f**) Top views of $bV_1$ (**d**) and $2_{ATP}V_1$ (**f**) as shown in **b**, which are superimposed at the 'bound' form onto that of $eV_1$ (grey). The bound AMP-PNP molecules are shown in space-filling representation, coloured in red.

## Results

**Structure of the 2AMP-PNP-bound $V_1$ complex.** We previously reported that nucleotide-free $EhV_1$ ($eV_1$) is composed of three different conformations of AB pairs: 'empty' ($A_OB_C$), 'bound' ($A_CB_{O'}$), and 'tight' ($A_{CR}B_{CR}$)[31] (Fig. 2a,b). A crystal structure of $bV_1$ corresponding to the catalytic dwell state was also obtained by soaking $eV_1$ crystals in mother liquor containing 200 µM AMP-PNP, which is sufficient to inhibit the ATP hydrolysis activity of purified $EhV_1$ (Supplementary Table 1). Two AMP-PNP:$Mg^{2+}$ molecules were bound to the binding sites of the 'bound' and 'tight' forms of $eV_1$, which did not cause further conformational changes of the A (Eh-A) and B (Eh-B) subunits, and the overall structure was similar to that of $eV_1$ (root mean square deviation (r.m.s.d. = 0.69 Å)[31] (Fig. 2c,d). These findings suggest that the binding affinities of AMP-PNP to the 'bound' and 'tight' forms are high, but that of the 'empty' form is low. In this study, in order to verify whether the third 'empty' form is able to bind AMP-PNP or not in the presence of a high concentration, we soaked the $eV_1$ crystals with 2 mM AMP-PNP, and solved the crystal structure (denoted $2_{ATP}V_1$) at a 2.7 Å resolution (Table 1). The structure showed two AMP-PNP

molecules bound in the 'bound' and 'tight' forms and was almost identical to that of $bV_1$ (r.m.s.d. = 0.51 Å) (ref. 31) (Fig. 2e,f). No electron density peak for AMP-PNP was found in the 'empty' form (Supplementary Fig. 2), indicating that it has a very low affinity for AMP-PNP.

**Structure of the 2ADP-bound $V_1$ complex.** Next, we soaked the crystals of $eV_1$ in 20 µM ADP, and the crystal structure (denoted $2_{ADP}V_1$) was solved at a resolution of 3.3 Å (Table 1). Two ADP:$Mg^{2+}$ molecules were bound to the 'bound' and 'tight' forms of $eV_1$, as in the case of $2_{ATP}V_1$ (Supplementary Fig. 3), and induced conformational changes with the crystal packing rearrangements (Supplementary Fig. 4). The structure of $2_{ADP}V_1$ was validated for possible model bias by generating omit maps of conformationally changed regions (Supplementary Fig. 5). ADP binding changed the structure of $eV_1$, but the crystal packing force might have the potential to distort the actual conformational changes for $2_{ADP}V_1$.

The structural differences between $eV_1$ and $2_{ADP}V_1$ that should have been induced by ADP binding are compared in Fig. 3 (see also Supplementary Movie 1). The $eV_1$ 'bound' form did not show a conformational change upon ADP binding (r.m.s.d. = 0.48 Å; Fig. 3c,d). However, the $eV_1$ 'tight' form changed to a more open conformation ($A_C$ from $A_{CR}$; $B_{C'}$ from $B_{CR}$) upon ADP binding (Fig. 3e,f). We designated the new ADP-bound $A_CB_{C'}$ pair of $2_{ADP}V_1$ as the 'ADP-bound' form. The γ-phosphate contained in AMP-PNP of the $bV_1$ 'tight' form interacted with the Lys238 residue of the P-loop ($P_i$-binding loop) and the Arg262 residue of the 'arm' region (fixed α-helix during the conformational changes: residues 261–275) in Eh-A, and the Arg350 residue (the so-called 'Arg-finger' in ATPases) in Eh-B to stabilize the 'tight' conformation[31] (Fig. 4b), thus preventing any further conformational change. In contrast, ADP, which does not contain γ-phosphate, interacted with these side chains by binding to β-phosphate (Fig. 4a and Supplementary Figs 6 and 7). These different binding contacts induced an apparent conformational change to the 'ADP-bound' form (Fig. 4a–c), which seems to be a more stable conformation for the ADP-binding mode than the 'tight' form. According to the observed conformational changes, the DF axis became tilted towards the 'ADP-bound' form to maintain the extensive protein–protein interactions between DF and the 'ADP-bound' form (Fig. 3b–h).

The last conformation of the AB pair ($eV_1$-'empty'), which did not bind to ADP, also showed a cooperative conformational change. Specifically, Eh-A ($A_O$) and Eh-B ($B_C$) of the 'empty' form were attracted to the DF axis and the 'ADP-bound' form, respectively (Fig. 3b,g,h and Supplementary Movie 1). The wider conformation of the resultant AB ($A_OB_{O''}$) pair was most similar to that of the $eA_3B_3$-'bindable' form ($A_OB_O$: ATP-accessible state) among all AB pairs (r.m.s.d. = 0.94 Å) (Supplementary Table 2), and was thus denoted a 'bindable-like' form. The structure at the nucleotide-binding site was also more similar to that of $eA_3B_3$-'bindable' (ATP-accessible state) than to that of $eV_1$-'empty' (ATP-unbound state). Similar to the 'bindable' form, the topology between the Arg-finger (Eh-B-Arg350) and Eh-A-Arg262 of the 'bindable-like' form was more open than that of the 'empty' form (Fig. 4d–f; green boxes). Therefore, the 'bindable-like' conformation seemed to be able to bind a nucleotide and probably changes to the 'bound' form, as observed for $eA_3B_3$-'bindable'[31]. Based on these findings, we inferred that the structure of $2_{ADP}V_1$ corresponds to the state of waiting for ATP binding (that is, the ATP-binding dwell) in the rotation.

**Structure of the 3ADP-bound $V_1$ complex.** Next, we soaked the $eV_1$ crystals in a high concentration (2 mM) of ADP to verify

**Table 1 | Data collection and refinement statistics of the V$_1$-ATPase.**

| Denoted as | 2$_{ATP}$V$_1$ | 2$_{ADP}$V$_1$ | 3$_{ADP}$V$_1$ | 0$_{Pi}$V$_1$:20 μM | 0$_{Pi}$V$_1$:200 μM | 1$_{Pi}$V$_1$ |
|---|---|---|---|---|---|---|
| *Crystallization condition* | | | | | | |
| Soaking with | 2 mM AMP-PNP 3 mM MgCl$_2$ | 20 μM ADP 3 mM MgSO$_4$ | 2 mM ADP 3 mM MgSO$_4$ | 20 μM P$_i$ 3 mM MgCl$_2$ | 200 μM P$_i$ 3 mM MgCl$_2$ | 2 mM P$_i$ 3 mM MgCl$_2$ |
| Soaking time | 6.5 h | 4.5 h | 4.5 h | 5.5 h | 5.0 h | 5.0 h |
| | | | | | | |
| *Data collection* | | | | | | |
| Beamline | PF BL1A | PF BL1A | PF BL1A | PF BL17A | PF BL1A | PF BL1A |
| Wavelength (Å) | 1.1000 | 1.1000 | 1.1000 | 0.9800 | 1.1000 | 1.1000 |
| Space group | P2$_1$2$_1$2$_1$ | P2$_1$2$_1$2$_1$ | P2$_1$2$_1$2$_1$ | P2$_1$2$_1$2$_1$ | P2$_1$2$_1$2$_1$ | P2$_1$2$_1$2$_1$ |
| Cell dimensions | | | | | | |
| $a, b, c$ (Å) | 128.3, 128.4, 226.9 | 127.4, 129.6, 237.2 | 121.7, 126.5, 225.3 | 128.5, 128.5, 226.5 | 127.9, 128.4, 226.7 | 128.2, 128.4, 228.0 |
| $\alpha, \beta, \gamma$ (°) | 90.0, 90.0, 90.0 | 90.0, 90.0, 90.0 | 90.0, 90.0, 90.0 | 90.0, 90.0, 90.0 | 90.0, 90.0, 90.0 | 90.0, 90.0, 90.0 |
| Resolution (Å) | 50–2.73 (2.89–2.73)* | 50–3.25 (3.45–3.25) | 50–3.02 (3.21–3.02) | 50–3.04 (3.23–3.04) | 50–2.84 (3.01–2.84) | 49.04–2.89 (2.99–2.89) |
| $R_{merge}$ | 0.170 (0.916) | 0.199 (1.064) | 0.221 (1.064) | 0.251 (0.873) | 0.222 (0.961) | 0.157 (0.824) |
| $I/\sigma I$ | 10.14 (1.87) | 10.33 (1.88) | 10.08 (1.92) | 8.79 (2.08) | 9.34 (1.99) | 12.5 (2.2) |
| Completeness (%) | 98.9 (94.4) | 99.4 (97.2) | 99.2 (95.4) | 99.7 (99.2) | 99.7 (98.7) | 99.9 (100) |
| Redundancy | 6.5 (6.3) | 6.7 (6.6) | 6.6 (6.5) | 6.3 (6.5) | 6.7 (6.8) | 6.6 (5.3) |
| | | | | | | |
| *Refinement* | | | | | | |
| Resolution (Å) | 50–2.73 | 50–3.25 | 50–3.02 | 48.93–3.04 | 50–2.84 | 49.04–2.89 |
| No. of reflections | 99,064 | 62,128 | 67,952 | 72,486 | 88,898 | 84,446 |
| $R_{work}/R_{free}$ (%) | 20.5/23.2 | 20.9/24.5 | 21.4/25.3 | 23.0/27.3 | 18.5/20.1 | 20.7/25.1 |
| No. of atoms | | | | | | |
| Protein | 26,653 | 25,976 | 26,554 | 26,389 | 26,414 | 26,309 |
| Ligand/ion | 137 | 68 | 173 | 44 | 80 | 43 |
| Water | 299 | 33 | 72 | 29 | 268 | 64 |
| B-factors | | | | | | |
| Protein | 55.37 | 83.02 | 55.82 | 49.78 | 44.90 | 64.84 |
| Ligand/ion | 47.81 | 57.03 | 57.91 | 56.10 | 50.85 | 70.55 |
| Water | 38.22 | 52.84 | 39.93 | 17.61 | 29.66 | 46.61 |
| r.m.s. deviations | | | | | | |
| Bond lengths (Å) | 0.004 | 0.003 | 0.003 | 0.002 | 0.002 | 0.003 |
| Bond angles (°) | 0.872 | 0.673 | 0.660 | 0.548 | 0.678 | 0.757 |
| PDB ID | – | 5KNB | 5KNC | – | – | 5KND |

All data sets were obtained from single crystal each. *Highest resolution shell is shown in parentheses.

nucleotide binding to the third 'bindable-like' form of 2$_{ADP}$V$_1$, and obtained the crystal structure (denoted 3$_{ADP}$V$_1$) at a resolution of 3.0 Å (Table 1). Three ADP:Mg$^{2+}$ molecules were bound at all three nucleotide-binding sites (Supplementary Fig. 8) and induced conformational changes with the crystal packing rearrangements (Fig. 5a, Supplementary Fig. 4, and Supplementary Movie 2). The structure was verified by generating omit maps of conformationally changed regions (Supplementary Fig. 5), although it is possible that the crystal packing force will distort the actual conformational changes for 3$_{ADP}$V$_1$. The structural differences between the 2ADP-bound (2$_{ADP}$V$_1$) and 3ADP-bound (3$_{ADP}$V$_1$) V$_1$ complexes, which are considered to be induced by ADP binding to the 'bindable-like' form of 2$_{ADP}$V$_1$, are compared in Fig. 5b–h (see also Supplementary Movie 3). 'Bindable-like' Eh-A (A$_{O''}$) changed to the half-closed conformation (denoted A$_{HC}$) upon ADP binding, whereas the B subunit (B$_{O''}$) did not (Fig. 5c,d). We thus designated the unique half-closed A$_{HC}$B$_{O''}$ pair of 3$_{ADP}$V$_1$ as the 'half-closed' form. A strong electron density peak for P$_i$ or SO$_4^{2-}$ (a P$_i$ analog) was observed at the nucleotide-binding site with ADP:Mg$^{2+}$ (Fig. 6a, and Supplementary Figs 7 and 8). We assigned this peak to SO$_4^{2-}$ because 3 mM MgSO$_4$, but not P$_i$, was contained in the crystallization condition. The adjacent 'bound' form was not affected by this conformational change to the 'half-closed' form. In contrast, the DF axis and 'ADP-bound' forms were slightly attracted to the A$_{HC}$ of the 'half-closed' form (Fig. 5b–h). The shifted 'ADP-bound' form was rather more similar to the observed 'tight' conformation (Fig. 5g,h).

Furthermore, the nucleotide-binding site was also more similar to that of the 'tight' form than to that of the 'ADP-bound' form (Fig. 6b–d, and Supplementary Fig. 7). We, therefore, designated this shifted 'ADP-bound' form of 3$_{ADP}$V$_1$ as the 'tight-like' form. The distances between the β-phosphate of ADP and the interacting residues in the 'tight-like' form were slightly longer than those in the 'ADP-bound' form (Supplementary Fig. 6), suggesting that the binding affinity for ADP of the 'tight-like' form is lower than that of the 'ADP-bound' form. Consequently, an ADP molecule will be easily released from the binding site. Therefore, we inferred that the structure of 3$_{ADP}$V$_1$ corresponds to the state of waiting for ADP release (that is, ADP-release dwell) in the rotation. ATP hydrolysis activity of purified EhV$_1$ was inhibited at a high (2 mM) concentration of ADP (Supplementary Table 1), which is significantly higher than the natural concentration in *E. hirae* cells. Therefore, the ADP-release dwell state might be a minor intermediate state, which might exist in the catalytic cycle with high [ADP] and low [ATP].

**Structure of the P$_i$-bound V$_1$ complex.** Next, we soaked the eV$_1$ crystals with 20 and 200 μM P$_i$, and solved the crystal structures at a resolution of 3.0 Å (0$_{Pi}$V$_1$:20 μM) and 2.8 Å (0$_{Pi}$V$_1$:200 μM), respectively (Table 1). However, no electron density peak for P$_i$, nor any conformational change was observed for either of the structures (Supplementary Fig. 9), and ATP hydrolysis activity of EhV$_1$ was not inhibited, even in the presence of 20 mM P$_i$ (Supplementary Table 1). These findings suggest that the binding

affinity for $P_i$ is lower than that of either AMP-PNP or ADP. We further soaked the crystals in a higher concentration (2 mM) of $P_i$, and the crystal structure (denoted $1_{Pi}V_1$) was determined at a resolution of 2.9 Å (Fig. 7 and Table 1). A $P_i$ molecule with $Mg^{2+}$ was found in the 'tight' form, which was fixed by the Arg-finger as observed for the binding of the γ-phosphate of AMP-PNP in $bV_1$ (Fig. 7 and Supplementary Fig. 7). Importantly, no conformational change was observed upon $P_i:Mg^{2+}$ binding (Fig. 7b); the overall structure was similar to those of $eV_1$ (r.m.s.d. = 0.46 Å)

and $2_{ATP}V_1$ (r.m.s.d. = 0.47 Å), but not to that of $2_{ADP}V_1$ (r.m.s.d. = 2.05 Å) or $3_{ADP}V_1$ (r.m.s.d. = 2.78 Å). Thus, the soaking of $eV_1$ crystals in $P_i$ did not induce conformational changes, as in the case of $2_{ATP}V_1$ after soaking with AMP-PNP. We also soaked the crystals of $eV_1$ in the mixture of various concentrations of AMP-PNP, ADP and/or $P_i$ to obtain other intermediate states. However, diffraction of these soaked crystals was not sufficient to solve the structure. Careful optimization of the ligand concentrations and crystal soaking times are necessary to improve the resolutions.

**Binding affinities of nucleotide to $V_1$ complex.** We performed isothermal titration calorimetry (ITC) experiments to estimate the binding affinities of AMP-PNP to nucleotide-free $EhV_1$. Exothermic reactions were observed upon the addition of AMP-PNP (Fig. 8a). The binding isotherm was saturated for titrations to an AMP-PNP/$EhV_1$ molar ratio of 2.2 (~14 μM AMP-PNP); no additional binding was observed for titrations up to the molar ratio of 1,400 (2.4 mM AMP-PNP) (Supplementary Fig. 10). The curve was fit by the two sets of sites model with the following parameters. The numbers ($n_1$ and $n_2$) of binding sites per $EhV_1$ were 0.68 and 0.77, respectively. The $K_{d1}$ and $K_{d2}$ values were 9.4 and 40 nM, respectively. The $\Delta H_1$ and $\Delta H_2$ values were −9.3 and −9.7 kcal mol$^{-1}$, respectively. The $\Delta S_1$ and $\Delta S_2$ values were 5.5 and 1.4 cal mol$^{-1}$ per degree, respectively. These ITC data suggested that the binding affinities of AMP-PNP to the 'bound' and 'tight' forms were both high, and that of the third 'empty' form was very low (<2 mM), corresponding to the structural findings described above.

Next, we quantified the binding affinities of ADP to nucleotide-free $EhV_1$ using ITC. The binding isotherm for ADP titration was remarkably different from that for AMP-PNP titration, and showed three distinct zones (Fig. 8b). The first zone, below an ADP/$EhV_1$ molar ratio of 2, was characterized by a continuous decrease in the exothermic signal. The second zone, between ADP/$EhV_1$ molar ratios of 2 and 2.8, exhibited the opposite trend, with an increase in the exothermic signal throughout the titration. Finally, in the third zone, the exothermic signal decreased as the ADP/$EhV_1$ molar ratios increased from 2.8 to 5.6 (a ratio at which saturation was reached), and no additional exothermic signal was observed for titrations up to an ADP/$EhV_1$ molar ratio of 1,400 (2.4 mM ADP) (Supplementary Fig. 10). Thus, the triphasic curve, which was likely to contain three different binding reactions, was analysed using the three sets of sites model[41] with the following parameters. The numbers ($n_1$, $n_2$ and $n_3$) of binding sites per $EhV_1$ were 1.4, 0.82 and 0.65, respectively. The $K_{d1}$, $K_{d2}$ and $K_{d3}$ values were 6.7 nM, 13 nM and 3.6 μM, respectively. The $\Delta H_1$, $\Delta H_2$ and $\Delta H_3$ values were −4.3, 1.3 and −10 kcal mol$^{-1}$, respectively. The $\Delta S_1$, $\Delta S_2$ and $\Delta S_3$ values were 23, 41 and -10 cal mol$^{-1}$ per degree, respectively. Interestingly, ADP binding to site-2 showed an endothermic reaction, whereas those to site-1 and site-3 involved exothermic reactions. This implies a dynamic structural change with ADP

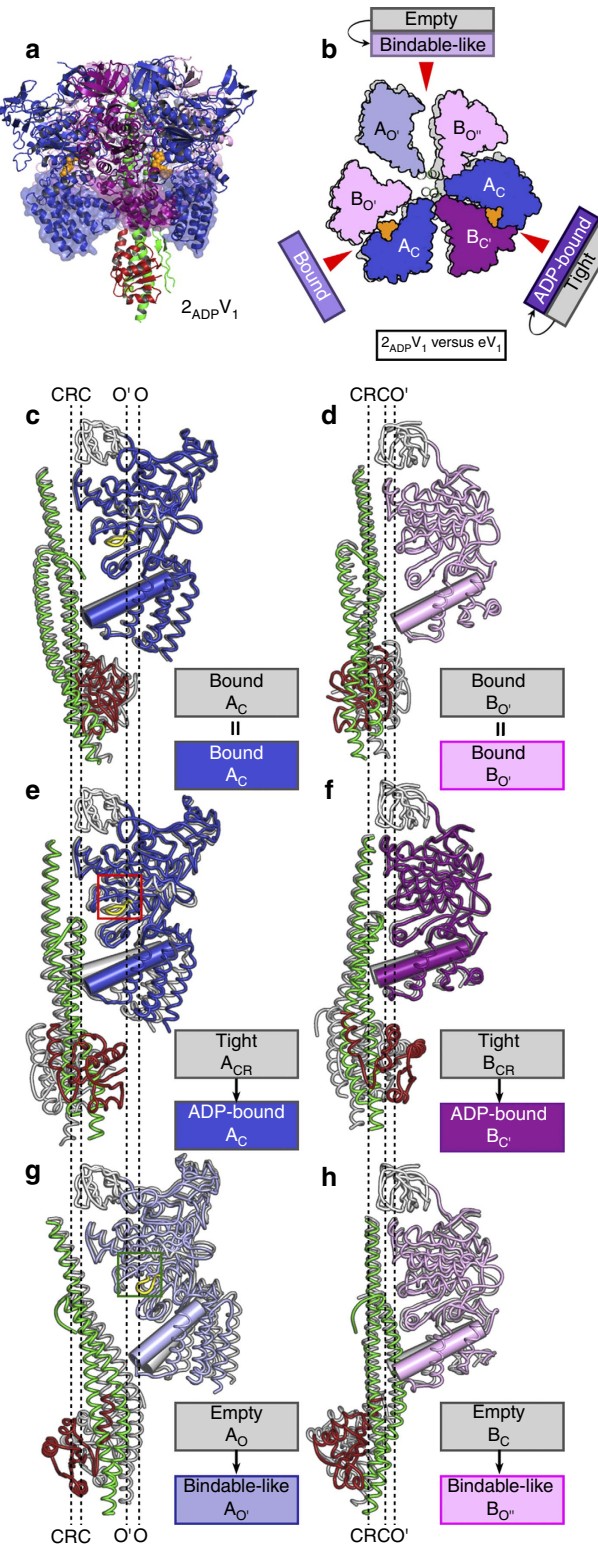

**Figure 3 | Structure of the 2ADP-bound $V_1$ complex ($2_{ADP}V_1$).** (**a**) Side view. (**b**) Top view of the C-terminal domain (transparent surface in **a**) from the cytoplasmic side in which the 'bound' conformation is superimposed onto that of $eV_1$ (grey). The bound ADP molecules are shown in space-filling representation and coloured orange. (**c–h**) Structural comparison of $2_{ADP}V_1$ and $eV_1$. $A_C$ (**c**), $B_{O'}$ (**d**), $A_C$ (**e**), $B_{C'}$ (**f**), $A_{O'}$ (**g**) and $B_{O''}$ (**h**) with DF in $2_{ADP}V_1$ are shown as side-view ribbon representations, which are superimposed at the N-terminal β-barrels (white) onto those of the corresponding subunits (grey) of $eV_1$. The 'P-loop' and 'arm' in Eh-A are shown in yellow and white, respectively. The longest α-helices of the C-terminal domains in Eh-A and Eh-B are shown as cylinders to clarify the structural differences.

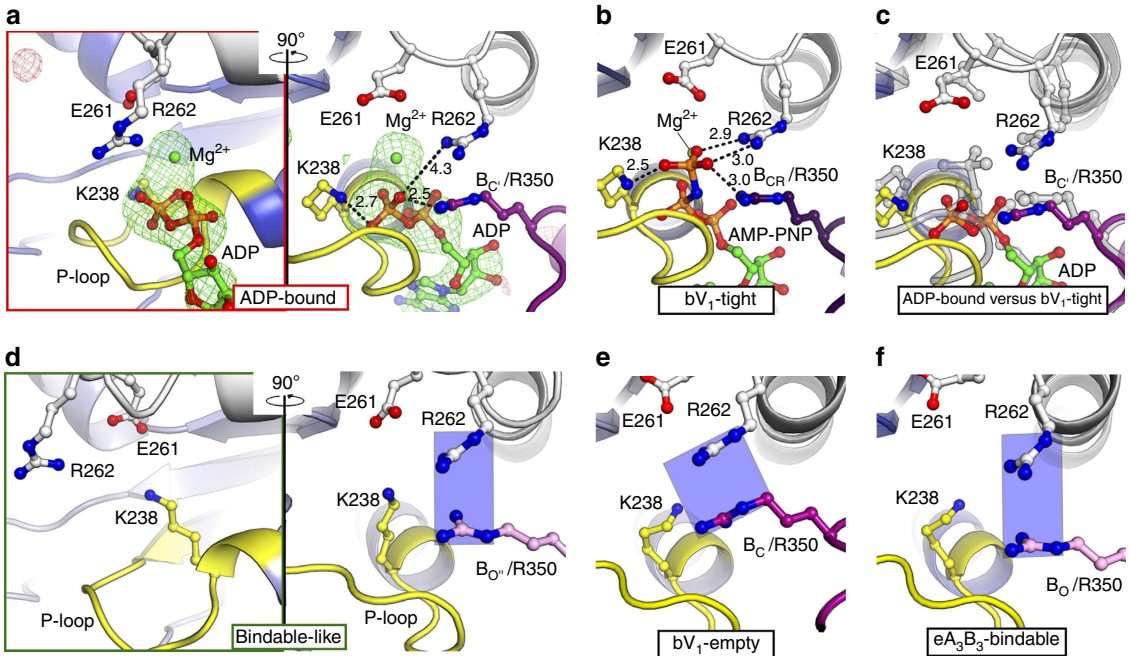

**Figure 4 | Nucleotide-binding sites of the 2ADP-bound V$_1$ complex (2$_{ADP}$V$_1$).** (**a**) Magnified view of the nucleotide-binding sites with conserved residues of the 'ADP-bound' form in 2$_{ADP}$V$_1$, corresponding to the red box of Fig. 3e. Right panels, A–B interfaces rotated 90° around a vertical axis from the left columns. The |Fo|-|Fc| maps calculated without ADP:Mg$^{2+}$ at the binding pockets contoured at 4.0 sigma are shown in red (negative) and green (positive). (**b,c**) The viewing position, colours and representations of the binding sites correspond to those described in the right panel of **a**. (**b**) bV$_1$-'tight'. Dotted lines indicate the distances (Å) between atoms. (**c**) **a** (right panel) is superimposed at the adenosine part onto that of **b** (shown in transparent grey). (**d**) Magnified nucleotide-binding site of the 'bindable-like' form (as in **a**), corresponding to the green box of Fig. 3g. (**e,f**) The viewing position, colours and representations of the binding sites correspond to those described in the right panel of **d**. (**e**) eV$_1$-'empty'. (**f**) eA$_3$B$_3$-'bindable'. Green boxes (**d–f**) show the topological locations of Eh-A-Arg262 and Eh-B-Arg350.

binding to site-2, corresponding to the structural findings that ADP binding to the 'tight' form induces conformational changes to the 'ADP-bound' form and the adjacent 'empty' form then changes to a 'bindable-like' form in a cooperative manner. According to this interpretation, site-1, -2 and -3 correspond to the 'bound', 'tight' and 'bindable-like' forms of EhV$_1$, respectively.

We also performed displacement ITC experiments of ADP-bound and AMP-PNP-bound EhV$_1$ by addition of AMP-PNP and ADP, respectively. The titration experiment of AMP-PNP into 3ADP-bound EhV$_1$ with 35 μM ADP (saturated concentration) showed that the exothermicity was remarkably lower than that into nucleotide-free EhV$_1$ (Fig. 8a,c). No noticeable exothermic signal was observed for titrations up to an AMP-PNP/EhV$_1$ molar ratio of 600 (2.4 mM AMP-PNP) (Supplementary Fig. 10). This suggests that AMP-PNP binding sites are already occupied by ADP in 3ADP-bound EhV$_1$, and these nucleotides competitively bind to EhV$_1$. In the competitive displacement experiment, the apparent $K_d$ values for AMP-PNP to 3ADP-bound EhV$_1$ were expected to be very high[42], and were actually estimated to be very high. The weak binding of AMP-PNP yields a nearly horizontal trace in binding isotherm whether or not the real displacement of nucleotides takes place[43]. Therefore, it is difficult to investigate the exchange reaction precisely from the ITC data. Similarly, the titration experiment of ADP into 2AMP-PNP-bound EhV$_1$ with 21 μM AMP-PNP (saturated concentration) also showed that the exothermicity was remarkably lower than that into nucleotide-free EhV$_1$ (Fig. 8b,d), and no noticeable exothermic signals were observed for titrations up to an ADP/EhV$_1$ molar ratio of 600 (2.4 mM ADP) (Supplementary Fig. 10); It is also predicted the exothermic signal should be very small because the apparent $K_d$ values for ADP to 2AMP-PNP-bound EhV$_1$ were estimated very high[42,43].

This finding suggests that ADP is not able to bind to the 'empty' form of 2$_{ATP}$V$_1$ owing to low affinity, as in the case of AMP-PNP. Therefore, the 'half-closed' form of 3$_{ADP}$V$_1$ seems to be obtained by ADP binding to the 'bindable-like' form of 2$_{ADP}$V$_1$, but not to the 'empty' form of eV$_1$ in the soaking experiment of ADP to eV$_1$ crystals, consistent with the structural findings in this study.

**Tryptophan fluorescence change of V$_1$ complex.** Tryptophan fluorescence is very sensitive to conformational changes in proteins[44]. In order to verify the conformational change induced by ADP binding, which was observed by X-ray crystallography of ADP-soaked crystals, we measured the tryptophan fluorescence of EhV$_1$ in the presence of AMP-PNP and/or ADP. Eh-A and Eh-B subunits of EhV$_1$ have 8 and 1 tryptophan residues, respectively. Emission spectra of the intrinsic tryptophan fluorescence of EhV$_1$ without added nucleotides showed a peak at 335 nm (Supplementary Fig. 11). The fluorescence intensity around 335 nm had distinct increase (2.3 ± 0.2 a.u.) by the addition of 500 nM AMP-PNP (a higher concentration than estimated $K_d$ values for AMP-PNP by ITC) (Fig. 8e, lane 1). However, the overall structure of 2$_{ATP}$V$_1$ was very similar to that of eV$_1$ (Fig. 2). Therefore, we attributed this change in intensity to a side-chain shift of the Trp248 residue near the P-loop from AMP-PNP binding, rather than overall conformational changes of EhV$_1$ (Supplementary Fig. 12). The fluorescence intensity was not affected by the re-addition of AMP-PNP (21 and 100 μM) (Fig. 8e, lane 2 and 3), consistent with the ITC data. On the other hand, the addition of 500 nM ADP (a higher concentration than the two $K_d$ values for ADP and lower than the third $K_d$ value for ADP) induced fluorescence change (3.1 ± 0.2 a.u.), which was higher than that of AMP-PNP (Fig. 8e, lane 4). This fluorescence increase is consistent with the conformational

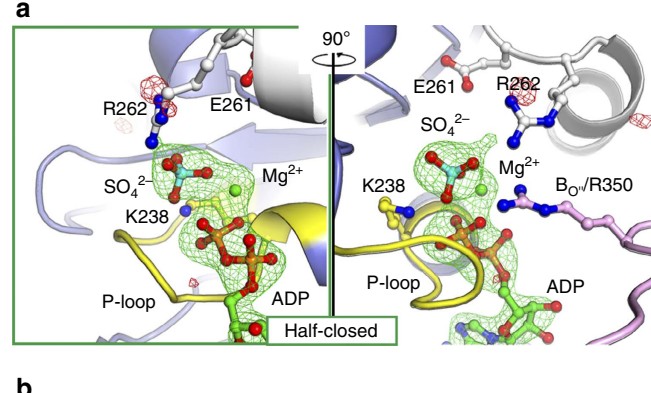

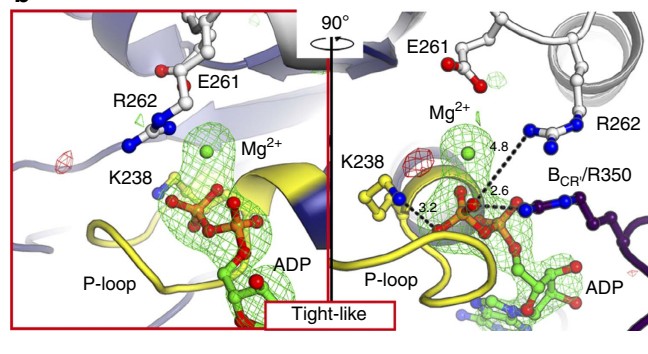

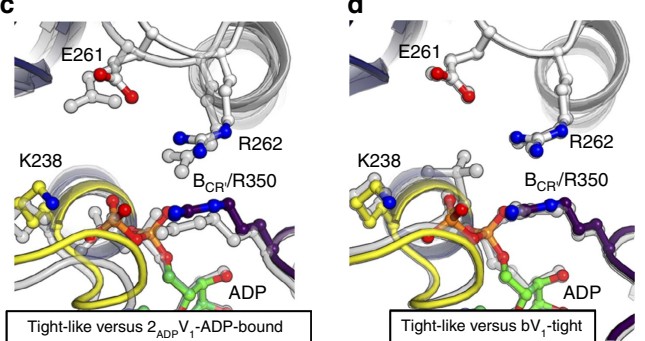

**Figure 6 | Nucleotide-binding sites of the 3ADP-bound V₁ complex (3$_{ADP}$V₁).** (**a,b**) Magnified nucleotide-binding site of the 'half-closed' (**a**) and 'tight-like' (**b**) forms in 3$_{ADP}$V₁ (as in Fig. 4a), corresponding to the green (Fig. 5c) and red (Fig. 5g) boxes, respectively. The |Fo|-|Fc| maps calculated without ADP:Mg$^{2+}$-SO$_4^{2-}$ (**a**) and ADP:Mg$^{2+}$ (**b**) at the binding pockets contoured at 4.0 sigma are shown in red (negative) and green (positive), respectively. SO$_4^{2-}$, cyan; Mg$^{2+}$, green. (**c,d**) The viewing position, colours and representations of the binding sites correspond to those in the right panel of **b**. The 'tight-like' form (**b**) in 3$_{ADP}$V₁ is superimposed at the adenosine part onto those (shown in transparent grey) of the 2$_{ADP}$V₁-'ADP-bound' (**c**) and bV₁-'tight' (**d**) forms.

**Figure 5 | Structure of the 3ADP-bound V₁ complex (3$_{ADP}$V₁).** (**a,b**) Top view of the C-terminal domain from the cytoplasmic side in which the 'bound' conformation is superimposed onto that (grey) of eV₁ (**a**) or 2$_{ADP}$V₁ (**b**). (**c–h**) Structural comparisons of 3$_{ADP}$V₁ and 2$_{ADP}$V₁. A$_{HC}$ (**c**), B$_{O''}$ (**d**), A$_C$ (**e**), B$_{O'}$ (**f**), A$_{CR'}$ (**g**) and B$_{CR'}$ (**h**) with DF in 3$_{ADP}$V₁ are shown as a side-view ribbon representations in which the N-terminal β-barrel (white) is superimposed onto those of the corresponding subunits (grey) of 2$_{ADP}$V₁. The colours and representations are the same as those described in Fig. 3c–h.

changes observed for the 2ADP-bound crystal structure (2$_{ADP}$V₁) (Supplementary Fig. 12). Addition of 35 μM ADP (a higher concentration than the third $K_d$ value for ADP) induced further changes in fluorescence intensity (Fig. 8e, lane 5), and the intensity was not affected by the addition of ADP at a higher concentration (that is, 100 μM ADP) (Fig. 8e, lane 6). This intensity change might correspond to the conformational changes to 3$_{ADP}$V₁ by ADP binding to the third 'bindable-like' form of 2$_{ADP}$V₁ (Supplementary Fig. 12).

We also performed competitive displacement experiments of ADP-bound and AMP-PNP-bound EhV₁ by addition of AMP-PNP and ADP, respectively. When 2 mM AMP-PNP was

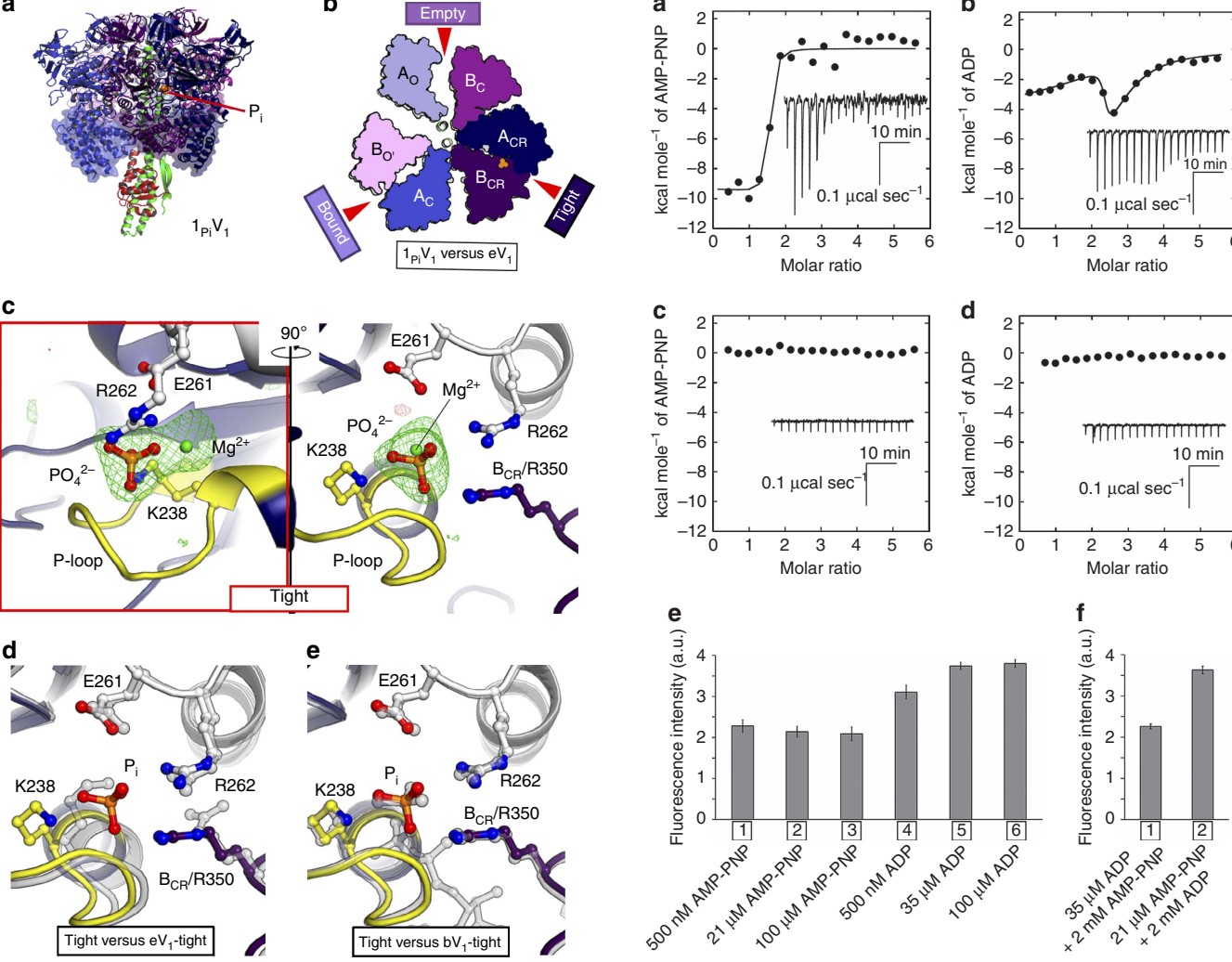

**Figure 7 | Structure of the P$_i$-bound V$_1$ complex (1$_{Pi}$V$_1$).** (a) Side view. (b) Top views of the C-terminal domain (transparent surface in **a**) from the cytoplasmic side in which the 'bound' form is superimposed onto that of eV$_1$ (grey). (c) Magnified nucleotide-binding site of the 'tight' form in 1$_{Pi}$V$_1$, as in Fig. 4a. The |Fo|-|Fc| maps calculated without P$_i$:Mg$^{2+}$ at the binding pockets contoured at 4.0 sigma are shown in red (negative) and green (positive), respectively. (**d**,**e**) The viewing position, colours and representations of the binding sites correspond to those in the right panel of **c**. The 'tight' form (**c**) in 1$_{Pi}$V$_1$, in which the Eh-A residues (67–593) are superimposed onto the same residues of the 'tight' form (shown in transparent grey) in eV$_1$ (**d**) and bV$_1$ (**e**), as in Fig. 4c. The bound P$_i$ molecule is depicted in stick format and coloured orange.

**Figure 8 | Biochemical properties of the *Enterococcus hirae* V$_1$ complex.** (**a**–**d**) Isothermal titration calorimetry (ITC) analysis. Nucleotides (200 μM) were injected into 7 μM EhV$_1$ at 25 °C. The integrated heat values from raw heats (inset) were plotted against the molar ratio of nucleotides to EhV$_1$ after subtraction of the nucleotide dilution heat values from the corresponding heat values of the EhV$_1$-nucleotide titration. (**a**,**b**) show the binding isotherm titrated to nucleotide-free EhV$_1$ with AMP-PNP (**a**) and ADP (**b**). The solid line represents the best fit to a binding model including the two sets of sites model for AMP-PNP (**a**) and the three sets of sites model for ADP (**b**). (**c**) shows the binding isotherm titrated to ADP-bound EhV$_1$ with AMP-PNP. (**d**) shows the binding isotherm titrated to AMP-PNP-bound EhV$_1$ with ADP. (**e**) Tryptophan fluorescence changes of nucleotide-free EhV$_1$ by addition of 500 nM (lane 1), 21 μM (lane 2) and 100 μM (lane 3) AMP-PNP and 500 nM (lane 4), 35 μM (lane 5) and 100 μM (lane 6) ADP. (**f**) Tryptophan fluorescence changes of 35 μM ADP-bound and 21 μM AMP-PNP-bound EhV$_1$ from nucleotide-free EhV$_1$ by addition of 2 mM AMP-PNP (lane 1) and 2 mM ADP (lane 2), respectively. The intensity was averaged between 330 and 340 nm. All data represent means ± standard estimated errors (s.e.m.) of three independent experiments.

added into 3ADP-bound EhV$_1$ pre-incubated with 35 μM ADP (saturated concentration), the fluorescence intensity decreased rapidly (Supplementary Fig. 11), and reached an equilibrium within 5 min (2.3 ± 0.1 a.u.; change from nucleotide-free EhV$_1$) (Fig. 8f, lane 1), which was very similar to that of AMP-PNP bound EhV$_1$ (Fig. 8e, lanes 1–3). Similarly, when 2 mM ADP was added into 2AMP-PNP-bound EhV$_1$ pre-incubated with 21 μM AMP-PNP (saturated concentration), the fluorescence intensity increased slowly to 3.6 ± 0.1 a.u. (change from nucleotide-free EhV$_1$) (Fig. 8f, lane 2 and Supplementary Fig. 11), which was very similar to that of 3ADP-bound EhV$_1$ (Fig. 8e, lanes 5 and 6). These findings suggest that EhV$_1$ is able to bind AMP-PNP and ADP at two or three binding sites competitively and to reversibly change the conformations.

## Discussion

We previously reported that the structure of bV$_1$ represents the catalytic dwell state (that is, a state of waiting for ATP hydrolysis) in which two ATP analogs (AMP-PNP) are bound, one in the 'bound' and the other in the 'tight' form. ATP hydrolysis is thought to occur in the 'tight' form due to induction caused by

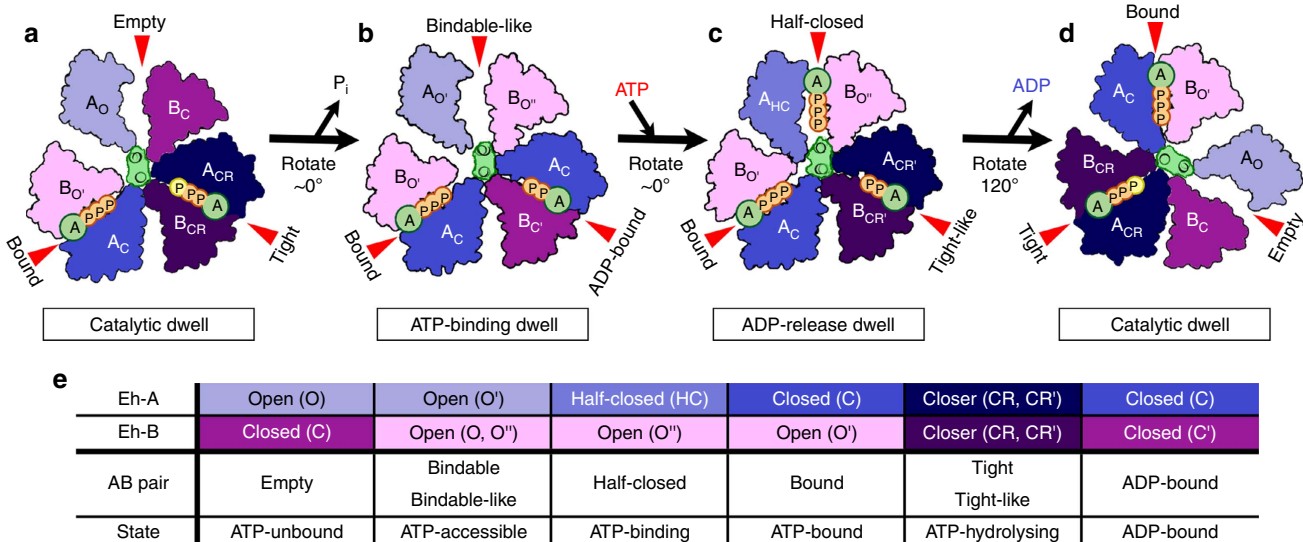

**Figure 9 | Proposed model of the rotation mechanism of *Enterococcus hirae* V₁-ATPase.** (**a**–**d**) The structure models are based on the crystal structures of $2_{ATP}V_1$ (catalytic dwell; **a**,**d**), $2_{ADP}V_1$ (ATP-binding dwell; **b**), and $3_{ADP}V_1$ (ADP-release dwell; **c**) determined in this study. ATP indicated as a yellow 'P' in (**a**) and (**d**) represents an ATP molecule that is committed to hydrolysis. (**e**) Correspondence table for all AB pairs observed in the crystal structures of the $A_3B_3$ and $V_1$ complexes. See text for additional details.

the approach of the Arg-finger[31]. However, the details of the reaction after hydrolysis remain unresolved. A new ATP molecule is unable to bind to the 'empty' form owing to its low affinity. In order for the reaction to continue, certain structural changes in the 'tight' form need to be induced via conversion to ADP and $P_i$. In this study, we solved the crystal structures of the 2ADP-bound $V_1$ complex ($2_{ADP}V_1$) by soaking $eV_1$ crystals in 20 μM ADP. The 'tight' form changed to the 'ADP-bound' form, and cooperatively induced conformational changes from the 'empty' to 'bindable-like' forms. In contrast, soaking with 20 and 200 μM $P_i$ did not produce an electron density peak for $P_i$ or any conformational change. Thus, ADP has a much higher binding affinity to the 'tight' form than does $P_i$. In the presence of 2 mM $P_i$, $P_i$:Mg$^{2+}$ was bound to the 'tight' form, but this binding did not induce conformational changes. Therefore, we concluded that $P_i$ is released first after ATP hydrolysis, which changes the conformation to the ATP-binding dwell state ($2_{ADP}V_1$). If ADP was released first, the conformational change required to continue the rotational reaction would not be induced, as observed for the $1_{Pi}V_1$ structure.

In contrast, the mammalian $F_1$-ATPase reaction model proposes that ADP is released first after hydrolysis (Fig. 1). This ADP-release-first order has also been observed for single-molecule manipulation of thermophilic $F_1$ (refs 19,20) and in molecular dynamics simulations based on the crystal structures of eukaryotic $F_1$ (refs 45–47). However, the release order is still uncertain owing to several inconsistent results among structural and single-molecule studies[10,21]. Regardless of the order, it is clear that the catalytic cycles of $F_1$ and $V_1$-ATPases differ substantially (see Fig. 1).

$EhV_1$ showed three 120° steps in a 360° rotation without apparent substeps in our single-molecule dynamics studies[32,33]. This rotational cycle pattern is similar to the rotational steps observed for *T. thermophilus* $V_1$ (ref. 25), whereas mammalian $F_1$ rotation involves two rotational substeps for 120° steps (Fig. 1). In this study, we obtained the crystal structures of three different states in the rotational cycle, corresponding to the catalytic dwell ($2_{ATP}V_1$), ATP-binding dwell ($2_{ADP}V_1$), and ADP-release dwell ($3_{ADP}V_1$) states. There are several potential explanations for why these states were not distinguished in our singe-molecule analysis.

First, there were clear structural differences between the catalytic dwell ($eV_1 = 2_{ATP}V_1$) and ATP-binding dwell ($2_{ADP}V_1$). The DF axis of $2_{ADP}V_1$ did not rotate significantly, but was instead tilted toward the 'ADP-bound' form owing to the conformational changes induced by the binding of ADP to the 'tight' form of $eV_1$ (see Supplementary Movie 1). Such a tilt of DF without apparent rotation would be difficult to detect using the single-molecule observations as an additional substep[32,33]. Second, there were clear structural differences between the ATP-binding dwell ($2_{ADP}V_1$) and the ADP-release dwell ($3_{ADP}V_1$). The DF axis of $3_{ADP}V_1$ was slightly bent towards the 'tight-like' form, but did not induce any rotational changes (see Supplementary Movie 3). This small shift in DF without apparent rotation would also be difficult to be detected using our single-molecule observations as additional substeps[32,33]. These findings suggested that $EhV_1$ exists in at least three dwell states in the 120° rotation without any rotational substeps. Thus, although the number of dwell states in $EhV_1$ and mammalian $F_1$ appears to be the same, these $V_1$ and $F_1$ motors show clear differences in the release order of cleavage products, rotational arrest points, dynamics and conformational changes.

Three-dimensional structures for three rotational states of the whole V-ATPase complex of *Saccharomyces cerevisiae* have been obtained by electron cryo-microscopy[48]. The samples for this analysis were obtained in the absence of nucleotides during the purification procedures. Therefore, the $V_1$ part of the three structures (PDB number: 3J9T, 3J9U and 3J9V) seemed to correspond to the nucleotide-free form. These three structures of yeast $V_1$ are comparable to those of $EhV_1$, and are the most similar to that of $eV_1$ (the nucleotide-free form of $EhV_1$ corresponding to the catalytic dwell state), although the tilts of these DF complexes are different (Supplementary Fig. 13). Thus, nucleotide-free $V_1$-ATPases seem to form the catalytic dwell state, rather than the ATP-binding dwell and ADP-release dwell states. Recently, the crystal structure of *S. cerevisiae* $V_1$-ATPase has been obtained at a 6.2 Å resolution[49]. This structure appears to be an inhibitory state wherein the subunit H inhibits the ATPase activity by stabilizing ADP binding to the catalytic site. We compared these structures of yeast $V_1$ and $EhV_1$, and found a lack of similarity between the yeast structure and $EhV_1$

structures ($2_{ATP}V_1$, $2_{ADP}V_1$, $3_{ADP}V_1$), suggesting that the yeast $V_1$ structure provides a unique view of an inhibitory state of a eukaryotic V-ATPase (Supplementary Fig. 13).

Finally, we propose a potential model to describe the rotation mechanism of *E. hirae* $V_1$-ATPase based on the observed crystal structures and single-molecule observations[32,33]. Figure 9 shows the 120° rotation model starting from the catalytic dwell (Fig. 9a), in which the surface structure of the C-terminal domain of 2AMP-PNP-bound $V_1$ ($2_{ATP}V_1$) is depicted, and the two ATP molecules are bound to the 'bound' and 'tight' forms (see Supplementary Fig. 14 and Supplementary Movie 4 for the 360° rotation model). The ATP that is tightly bound to the 'tight' form is hydrolysed to produce ADP and $P_i$[31]. The $P_i$ molecule, which has a lower affinity than ADP, is released, inducing a change from the 'tight' to 'ADP-bound' form. Consequently, the DF axis tilts toward the 'ADP-bound' form, but this does not induce a rotational event of the DF axis. The adjacent 'empty' form (ATP-unbound state) then changes to a 'bindable-like' form (ATP-accessible state) in a cooperative manner (Fig. 9b: ATP-binding dwell). Next, a new ATP molecule binds to the 'bindable-like' form, which induces a conformational change to the 'bound' form, thereby releasing the bound ADP from the 'ADP-bound' form. If the ADP stays in the 'ADP-bound' form, the 'bindable-like' form will become 'half-closed' due to ATP binding (more specifically, due to ADP:$Mg^{2+}$ binding with $SO_4^{2-}$, as shown in Fig. 6a), which is accompanied by a small shift of the DF axis, but no apparent rotational substep. Consequently, the adjacent 'ADP-bound' form cooperatively returns to the 'tight-like' conformation, and the binding affinity for ADP is reduced, as described above (Fig. 9c: ADP-release dwell). Then, ADP is released from the 'tight-like' form, and the 'half-closed' form is converted to the 'bound' conformation. Following these conformational changes, the DF axis rotates 120° with a torque of ~25 pNnm (ref. 33), and conformational changes from 'tight-like' to 'empty' and from 'bound' to 'tight' occur as a result of protein–protein interactions with the DF axis[31]. Finally, the enzyme resumes its initial catalytic dwell state, shown in Fig. 9a (Fig. 9d). Thus, the $V_1$ motor achieves its rotational dynamics via several conformational changes that are generated by the binding of ATP, release of $P_i$ and ADP, and cooperative coupling among these conformational changes. We will refine our model further by performing a combination of structural, single-molecule and computational analyses to fully understand the operational mechanism of $EhV_1$.

## Methods

**Protein preparation.** An *Escherichia coli* cell-free protein expression system was used to synthesize the $A_3B_3$ and DF complexes using a mixture of plasmids harbouring the corresponding genes with a modified natural poly-histidine (MKDHLIHNHHKHEHAHAEH) affinity tag, tobacco etch virus cleavage site (EHLYFQG) and linker (SSGSSG) sequences at the N terminus[29–31]. The reacted cell-free lysate was loaded onto a HisTrap HP column (GE Healthcare, Little Chalfont, UK) equilibrated with buffer-A (50 mM Tris-HCl, 750 mM NaCl, 5 mM 2-mercaptoethanol, and 10 mM imidazole; pH 8.0), and bound proteins were eluted with buffer-B (50 mM Tris-HCl, 300 mM NaCl, 5 mM 2-mercaptoethanol, and 500 mM imidazole; pH 8.0). The sample buffer was exchanged to buffer-A using a HiPrep 26/10 desalting column (GE Healthcare, Little Chalfont, UK). To obtain non-tagged samples, the proteins were treated with tobacco etch virus protease at 4 °C for 12 h. The reaction solution was loaded onto a HisTrap HP column again, and the flow-through fractions containing the non-tagged proteins were pooled. The protein samples were dialysed against buffer-C (50 mM Tris-HCl, 10 mM NaCl, 5 mM 2-mercaptoethanol; pH 8.5), loaded onto a HiTrap Q HP column (GE Healthcare, Little Chalfont, UK) equilibrated with buffer-C and eluted with a linear gradient of 10–1,000 mM NaCl. Finally, the concentrated samples with an Amicon Ultra 30 K unit (Merck Millipore, Darmstadt, Germany) were loaded onto a HiLoad 16/60 Superdex 200 pg column (GE Healthcare, Little Chalfont, UK) equilibrated with buffer-D (20 mM Tris-HCl, 150 mM NaCl, and 2 mM dithiothreitol; pH 8.0) and eluted using buffer-D. The purified $A_3B_3$ and DF complexes were concentrated with an Amicon Ultra 30 K unit.

$V_1$-ATPase ($A_3B_3DF$) was reconstituted and purified as follows: purified $A_3B_3$ and DF in buffer-D were mixed in a 1:5 molar ratio with the addition of MES (100 mM final concentration; pH 6.0) and incubated with 0.2 mM AMP-PNP and 5 mM $MgSO_4$ for 1 h. Reconstituted $V_1$-ATPase was purified using a HiLoad 16/60 Superdex 200 pg column equilibrated with buffer-E (20 mM MES, 10% glycerol, 100 mM NaCl, 5 mM $MgSO_4$, and 2 mM dithiothreitol; pH 6.5). Purified complexes were concentrated with an Amicon Ultra 30 K unit.

**Crystallization.** Crystals of nucleotide-free $A_3B_3DF$ ($eV_1$) were obtained by mixing 0.1 µl of 8 mg ml$^{-1}$ purified $V_1$-ATPase in buffer-E (see previous section) with 0.1 µl of reservoir solution (0.1 M Bis-Tris propane (pH 6.5–7.5), 20–22% polyethylene glycol (PEG)-3350, and 0.2 M NaF), using the sitting-drop vapour diffusion method at 296 K. The crystals were soaked in the following conditions (i–vi), mounted on cryo-loops (Hampton Research, Aliso Viejo, CA, USA), flash-cooled, and stored in liquid nitrogen.

  i. $V_1$-ATPase soaked with 2 mM AMP-PNP ($2_{ATP}V_1$): The $eV_1$ crystals were soaked for 6.5 h in 0.1 M Bis-Tris propane (pH 6.5), 21% PEG-3350, 2 mM AMP-PNP, 3 mM $MgCl_2$, 0.28 M NaCl and 20% glycerol.
  ii. $V_1$-ATPase soaked with 20 µM ADP ($2_{ADP}V_1$): The $eV_1$ crystals were soaked for 4.5 h in 0.1 M Bis-Tris propane (pH 6.5), 21% PEG-3350, 20 µM ADP, 3 mM $MgSO_4$, 0.28 M NaCl, and 20% glycerol.
  iii. $V_1$-ATPase soaked with 2 mM ADP ($3_{ADP}V_1$): The $eV_1$ crystals were soaked for 4.5 h in 0.1 M Bis-Tris propane (pH 6.5), 21% PEG-3350, 2 mM ADP, 3 mM $MgSO_4$, 0.28 M NaCl and 20% glycerol.
  iv. $V_1$-ATPase soaked with 20 µM $P_i$ ($0_{Pi}V_1$:20 µM): The $eV_1$ crystals were soaked for 5.5 h in 0.1 M Bis-Tris propane (pH 7.5), 21% PEG-3350, 20 µM sodium phosphate, 3 mM $MgCl_2$, 0.28 M NaCl and 20% glycerol.
  v. $V_1$-ATPase soaked with 200 µM $P_i$ ($0_{Pi}V_1$:200 µM): The $eV_1$ crystals were soaked for 5.0 h in 0.1 M Bis-Tris propane (pH 6.5), 21% PEG-3350, 200 µM sodium phosphate, 3 mM $MgCl_2$, 0.28 M NaCl and 20% glycerol.
  vi. $V_1$-ATPase soaked with 2 mM $P_i$ ($1_{Pi}V_1$): The $eV_1$ crystals were soaked for 5.0 h in 0.1 M Bis-Tris propane (pH 6.5), 21% PEG-3350, 2 mM sodium phosphate, 3 mM $MgCl_2$, 0.28 M NaCl and 20% glycerol.

**Structure determination.** All X-ray diffraction data were collected from a single crystal at a cryogenic temperature (100 K) at the Photon Factory (Tsukuba, Japan). The collected data were processed using XDS[50] or HKL2000 software (HKL Research, Inc., Charlottesville, VA, USA). The structures were solved by molecular replacement with Phaser[51] or MOLREP[52], using the crystal structures of $bV_1$, $eV_1$, $2_{ADP}V_1$, $bV_1$, $1_{Pi}V_1$ and $bV_1$ as a search model for $2_{ATP}V_1$, $2_{ADP}V_1$, $3_{ADP}V_1$, $0_{Pi}V_1$:20 µM, $0_{Pi}V_1$:200 µM and $1_{Pi}V_1$, respectively. The atomic models were built using Coot[53], cross-validated by making various omit maps to minimize model bias, and iteratively refined using REFMAC5 (ref. 54) and Phenix[55]. TLS (Translation/Libration/Screw) refinement was performed in late stages of refinement. The refined structures were validated with RAMPAGE[56]. For the structures of $2_{ATP}V_1$, $2_{ADP}V_1$, $3_{ADP}V_1$, $0_{Pi}V_1$:20 µM, $0_{Pi}V_1$:200 µM and $1_{Pi}V_1$, 99.9, 99.9, 99.9, 100, 99.9 and 99.9% of the residues, respectively, were in favoured or allowed regions based on a Ramachandran analysis. The crystallographic and refinement statistics are summarized in Table 1. All r.m.s.d. values were calculated using Cα atoms. The r.m.s.d. values for the superimpositions for each AB pair in the crystal structures are listed in Supplementary Table 2. Figures were prepared using PyMOL (The PyMOL Molecular Graphics System, Version 1.3, Schrodinger, LLC, New York, NY, USA).

**Measurement of ATPase activity and protein concentrations.** ATPase activity of the purified $V_1$-ATPase in the presence of AMP-PNP or ADP was measured by the colorimetric method using molybdic acid[57,58]. The reaction was initiated by the addition of 1 mM ATP, after a 10 min pre-incubation with various concentrations of AMP-PNP or ADP, and terminated by the addition of 10% sodium dodecyl sulphate. The initial rate of the ATPase reaction at 23 °C was determined within 4 min, and the measurement was repeated three times. ATPase activities of the purified $V_1$-ATPase in the presence of various concentrations of sodium phosphate were measured using an ATP regenerating system[31,59]. ATP hydrolysis rates at 23 °C were determined in terms of the rate of NADH oxidation, which was measured as a decrease in absorbance at 340 nm, and the measurement was repeated three times. Protein concentrations were determined using Pierce BCA Protein Assay Kit (Thermo Fisher Scientific, Inc., Waltham, MA, USA) with bovine serum albumin as the standard.

**Isothermal titration calorimetry (ITC).** $V_1$-ATPase was prepared by mixing 12 µM $A_3B_3$ and 60 µM DF in 900 µl of buffer-D, and the suspended sample buffer was replaced with buffer-F (100 mM Tris-HCl, 100 mM NaCl, and 5 mM $MgSO_4$; pH 7.5) using Spectra/Por 3 Dialysis Tubing (Spectrum Laboratories, Inc., Rancho Dominguez, CA, USA). ITC experiments were performed using the MicroCal iTC200 calorimeter (Malvern Instruments Ltd., Malvern, Worcestershire, UK), and the samples (7 µM) with/without 21 µM ANP-PNP or 35 µM ADP were loaded into the sample cell. Either 200 µM AMP-PNP or ADP in buffer-F was injected into the sample cell at 25 °C using one initial injection of 1.0 µl followed by 18

injections of 2.0 μl. Binding data were fitted to the two sets of sites model using Origin 7.0 (MicroCal) or the three sets of sites model using MATLAB[41].

**Tryptophan fluorescence.** $V_1$-ATPase was prepared by mixing 100 nM $A_3B_3$ and 500 nM DF in buffer-F in 1.2 ml. Fluorescence experiments were performed using the FP-6500 spectrofluorometer (JASCO, Tokyo, Japan) at 25 °C. Fluorescence spectra were recorded with excitation at 300 nm (slit width 1 nm) and emission between 310 and 450 nm (slit width 20 nm). Time courses of exchange reactions of AMP-PNP and ADP were measured every 0.5 s at 335 nm with excitation at 300 nm. Time courses were averaged for 20 data points around each point.

**Data availability.** Coordinates and structure factors for the ADP- and $P_i$-bound $V_1$-ATPase complexes have been deposited in the Protein Data Bank under the accession codes 5KNB (doi: 10.2210/pdb5knb/pdb; $2_{ADP}V_1$ at 3.3 Å) (ref. 60), 5KNC (doi: 10.2210/pdb5knc/pdb; $3_{ADP}V_1$ at 3.0 Å)[61], and 5KND (doi: 10.2210/pdb5knd/pdb; $1_{Pi}V_1$ at 2.9 Å)[62]. The authors declare that all other relevant data supporting the findings of this study are available within the article and its Supplementary Information files.

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

## Acknowledgements

This paper is dedicated to Late Prof Yoshimi Kakinuma (he died at the age of 64 in January 2016) as a memorial for his great contribution to the research of V-type ATPases. He studied $Na^+$-translocating ATPase in *E. hirae* and found it to be a bacterial V-type ATPase. With his support, we succeeded in obtaining crystal structures of the $V_1$-ATPase as described in this paper and are now close to understanding its bioenergy transduction mechanism. We also thank Drs Satoshi Arai, Shinya Saijo, Ryota Iino, Hiroshi Ueno, Yoshihiro Minagawa, Mitsunori Ikeguchi and Keitaro Yamashita for helpful suggestions on this study. The synchrotron radiation experiments were performed at Photon Factory (proposals 2011S2-005, 2012G-132, 2013R-22, 2014R-51). We also thank the beamline staff at BL1A and BL17A of Photon Factory (Tsukuba, Japan) for help during data collection. This work was supported in part by Grants-in-aid for Scientific Research (26291009 to T.M., 13J04652 to K. Suzuki, JP16H00811 to E.M.) from the Ministry of Education, Culture, Sports, Science and Technology, Japan (MEXT), and by the Platform Project for Supporting in Drug Discovery and Life Science Research (Platform for Drug Discovery, Informatics and Structural Life Science) from MEXT and Japan Agency for Medical Research and development (AMED).

## Author contributions

T.M. designed the study. Y.K., Y.I.-K, M.S. and S.Y. prepared the proteins. K. Suzuki crystallized the proteins. K. Suzuki and K.M. collected X-ray data. K. Suzuki, K.M. and F.L.I. processed and refined X-ray data. K. Suzuki, S.M, K. Shimono and E.M. performed functional analysis. K. Suzuki, K. Shimono, E.M., I.Y. and T.M. analysed the results. K. Suzuki prepared figures and movies. T.M. wrote the paper. All authors discussed the results and commented on the manuscript.

## Additional information

**Competing financial interests:** The authors declare no competing financial interests.

