## [Peer review file · Nature Communications]

Reviewer #1 (Remarks to the Author):

Suzuki et al. & Murata present a thorough, crystallographic report of the V1-ATPase based on soaking studies of nucleotide-free V1-ATPase crystals with ADP, AMPPNP and Pi. They find structural changes associated with ADP binding. Soaking with AMPPNP or Pi at low, medium and high concentrations also reveal binding events, but not associated with similar conformational changes. They interpret these data in context of the available literature and reach a discussion of dwell states in the rotary mechanism of V-ATPases and a revised model of the rotary mechanism

1) The data certainly appears to bear merit of new insight into the mechanism of V-ATPases, but the study seems too narrow and is based on the soaking studies alone, i.e. no complementary data to support the observations. Crystal soaking is a very useful approach when practically possible and validated, but one must also keep in mind that a crystal is a restrictive environment for conformational changes, which may not complete or perhaps be misleading, or not be observed at all due to crystal packing interactions overruling the ligand-induced changes. It would seem necessary to include other binding data than those observed in the soaked crystals, such as isothermal calorimetry, thermophoresis or surface plasmon resonance. Probing for conformational changes by e.g. Trp fluorescence may also be very helpful to qualify and validate the crystal observations.

2) The manuscript is generally written in a fine command of English, but the flow of observations, interpretations, rationales and discussion that build up the dissemination are very difficult to follow for a non-expert in the field, which would be the typical readership aimed for by this submission. A thorough revision is required, and should as mentioned also above include additional data to support the ligand-binding observations and induced changes.

Small remarks:

3) The opening sentence of the abstract (V-ATPases as attractive drug target) is over-exposed and off-topic as this point is not addressed by the findings in the paper and does not serve to motivate the study otherwise. It should either be moderated, left out or addressed in the study

4) "soaking [...] crystals into solution containing ..." (abstract and other places) sounds a bit odd and leaves the impression that crystals are dissolved

5) "inside of heaxogonally arranged [...] A3B3 and alpha3beta3...." (page 3) would be pseudo-hexagonally

6) Mixed soaks, e.g. ADP, Pi and AMPPNP altogether, would be very helpful to explore further the consistent sets of conformational changes that are relevant to the functional cycle.

Reviewer #2 (Remarks to the Author):

The paper by Suzuki et al details a new crystal structure of the "V1" domain in and apparent ADP release dwell. A number of structures are reported based on crystal soaking experiments which reveal new insights into the rotational mechanism for the rotary ATPases. This work will be of interest to many people in the rotary ATPase field along with those interested in molecular motors.

With regards the work there are a number of key points I would like to raise.

1) The first is the insistence on the authors calling this the V1 domain. There have been number of structural and biochemical studies which have shown a distinct difference between the F-ATPase, A-ATPase and V-ATPase. The structure studied here is of the A-ATPase family which are structurally and functionally distinct from the V-ATPase family. Indeed the paper makes no mention of the A-ATPase family. This structure being from the A-ATPase alone does not in my eyes diminish the importance, I think it still shows new novel insights, but there must be clarity in the field as to which system is which. If the authors believe strongly that this is a V not A-ATPase then this must be addressed fully within the paper for the readers to make up their own mind. They must also for clarity discuss the F/V and A-ATPase systems rather than just the F and V.

2) Although there appears to be a distinct difference between this crystal form and others

reported, this is still based on soaking and not co-crystallisation experiments. As such crystallisation artefacts maybe present. For example the authors talk about rotation of the DF axle, however is this compromised by the crystal contacts? It would be good to see the contacts in a sup. Figure. Its not clear how you can pack the "V1" domain crystals without interactions at the DF axle. Related to this the ADP structure is generated by the addition of 2mM ADP, this is significantly higher than a natural concentration. How representative is this of the natural state. It would be good for the authors to state the calculated affinity for their "V1" domain to ATP and ADP to give the readers an insight into how much more is added for co-crystallisation. Also How did the authors check for bias within their structures given the molecular replacement approach on modest resolution data. I think they should state for clarity how they determined the level of bias, i.e. did they use poly alanine models and check for side chain density, use omitted parts of the structure to check the information was coming back etc... By using the same protein for molecular replacement it can be easy to obtain bias, although the larger movements may still be apparent more subtle changes maybe lost.

3) It seems strange that the authors do not refer to the recent cryo-EM structures, especially that by John Rubinstein of the V-ATPase in three distinct states. How well do their structures compare? This would be a good way to back up the validity of a co-crystal structure on just part of a complex. It has also been shown in Rawson et al. that there is a significant amount of movement required to fit the central axle from the crystal structures into the cryo-EM map, reflective of some of the problems of crystallising an isolated domain rather than the whole complex. Of course the EM maps are not as high a resolution as shown here which is why comparing some of the EM data with the crystal structures could be a powerful approach.

The above points do raise questions on the reliability of these results and should be addressed fully by the authors.

I do not believe any additional experiments are required but I do think that the authors must review there current data in light of the points raised. With regards improvements;

Minor points;

1) I can not find a legend for table 1.

2) I see no need for or a top and bottom view in Figure 1. One would suffice and also cytoplasmic and luminal, or cytoplasmic and extracellular maybe more informative. Using N-terminal and C-terminals side in the figure legend is likely to be of little use to a wider audience.

3) Both the abstract and introduction start by discussing the role of the V-ATPase in disease and its potential as a therapeutic target. However, this is not mentioned again in the manuscript and its unclear how this work is informing drug design. I would remove this.

4) Introduction, line 4, the authors compare the F and V-ATPase structures but fail to mention which subunits are from which complex. The sentence is also overlong.

5) r.m.s.d. values are given throughout the text but are these of the Ca atoms or all atoms. Also given the resolution of $\sim 3\text{\AA}$, can r.m.s.d. values to 3 d.p. be justified?

6) Figure 3 legend the ADP is referred to as orange spheres, I would just state colored orange as there is they are not very spherical.

7) In Figure 4 the distances are shown but depending on the reproduction size of the figure I worry they will be unreadable.

8) Figure 5 states "tight form (color) in ..." Which color?

9) Also in the legend for figure 5 it states ADP is depicted as an orange stick, I think it would be

better to refer to this as "in stick format, colored orange"

10) In the methods, crystallisation was carried out in "buffer b" what is buffer b? If this is a brand name from a certain supplier then please note this otherwise I am not sure how these experiments could be repeated.

11) I find the colour schemes at times confusing with the use of light and dark colours especially in the side views and superimpositions. For example Figure 2 a, c and e all the A and B domains look the same colour. The black and dark purple colours also provide little contrast. I would recommend more distinct colours to make it clearer to the reader.

The references seem appropriate, however as discussed above I think a few references on the rotary ATPase family F/A & V would be beneficial (there are a number out there now with recent structures of all three).

In the main the clarity is good and the schematics of the proposed new catalytic cycle useful. It maybe difficult for people outside the direct field to follow but I am not sure if this is of much concern.

Title: Crystal structures of the ATP-binding and ADP-release dwells of the V₁ rotary motor

Authors: Kano Suzuki, Kenji Mizutani, Shintaro Maruyama, Kazumi Shimono, Fabiana L. Imai, Yoshimi Kakinuma, Yoshiko Ishizuka-Katsura, Mikako Shirouzu, Shigeyuki Yokoyama, Ichiro Yamato, and Takeshi Murata

Response to Referees comments

Referee #1:

Comment 1-1: The data certainly appears to bear merit of new insight into the mechanism of V-ATPases, but the study seems too narrow and is based on the soaking studies alone, i.e. no complementary data to support the observations. Crystal soaking is a very useful approach when practically possible and validated, but one must also keep in mind that a crystal is a restrictive environment for conformational changes, which may not complete or perhaps be misleading, or not be observed at all due to crystal packing interactions overruling the ligand-induced changes. It would seem necessary to include other binding data than those observed in the soaked crystals, such as isothermal calorimetry, thermophoresis or surface plasmon resonance. Probing for conformational changes by e.g. Trp fluorescence may also be very helpful to qualify and validate the crystal observations.

A: Thank you for your advice. In response to this suggestion, we have performed isothermal titration calorimetry (ITC) experiments to estimate the binding properties of AMP-PNP and ADP, and have revised the text to include the ITC binding data (page 8, line 6; page 13, line 12; Figure 3; Supplementary Figure 3). We also performed a Trp fluorescence experiment to validate the crystal observations. However, we were not able to detect reliable fluorescence changes in response to the addition of nucleotides owing to

high background Trp fluorescence attributed to 27 Trp residues in EhV₁ (Eh-A, 8 residues; Eh-B, 1 residue).

Comment 1-2: The manuscript is generally written in a fine command of English, but the flow of observations, interpretations, rationales and discussion that build up the dissemination are very difficult to follow for a non-expert in the field, which would be the the typical readership aimed for by this submission. A thorough revision is required, and should as mentioned also above include additional data to support the ligand-binding observations and induced changes.

A: We agree with your assessment of the manuscript structure and have revised the text extensively for readability. We have also added data to support the ligand-binding observations as mentioned in our response to comment 1-1.

Comment 1-3: The opening sentence of the abstract (V-ATPases as attractive drug target) is over-exposed and off-topic as this point is not addressed by the findings in the paper and does not serve to motivate the study otherwise. It should either be moderated, left out or addressed in the study.

A: We agree that the implications for drug target identification were not described in sufficient detail and did not serve to motivate the study. Accordingly, we have deleted the original opening sentence in the revised abstract.

Comment 1-4: "soaking [...] crystals into solution containing ..." (abstract and other places) sounds a bit odd and leaves the impression that crystals are dissolved.

A: We have corrected these phrases in the revised manuscript to avoid suggesting that the crystals were dissolved.

Comment 1-5: "inside of heaxogonally arranged [...] A3B3 and alpha3beta3...." (page 3) would be pseudo-hexagonally

A: Thank you for pointing out this error. It has been corrected in the revised manuscript.

Comment 1-6: Mixed soaks, e.g. ADP, Pi and AMPPNP altogether, would be very helpful to explore further the consistent sets of conformational changes that are relevant to the functional cycle.

A: We agree with this suggestion. We attempted mixed-soaking experiments. However, we were unable to obtain the structures owing to the low resolution of the soaked crystals (worse than 6 Å) so far, which probably reflects the heterogeneity of the bound substrates in the crystals.

Referee #2:

Comment 2-1: The first is the insistence on the authors calling this the V1 domain. There have been number of structural and biochemical studies which have shown a distinct difference between the F-ATPase, A-ATPase and V-ATPase. The structure studied here is of the A-ATPase family which are structurally and functionally distinct from the V-ATPase family. Indeed the paper makes no mention of the A-ATPase family. This structure being from the A-ATPase alone does not in my eyes diminish the importance, I think it still shows new novel insights, but there must be clarity in the field as to which system is which. If the authors believe strongly that this is a V not A-ATPase then this must be addressed fully within the paper for the readers to make up their own mind. They must also for clarity discuss the F/V and A-ATPase systems rather than just the F and V.

A: Thank you for raising this point regarding the differences between ATPase types. In the revised text, we have clarified the differences between F-, A-, and V-ATPases and justified referring to the structure as a homologue of eukaryotic V-ATPase (page 4, line 2; page 5, line 11).

Comment 2-2: Although there appears to be a distinct difference between this crystal form and others reported, this is still based on soaking and not co-crystallisation experiments. As such crystallisation artefacts maybe present. For example the authors talk about rotation of the DF axle, however is this compromised by the crystal contacts? It would be good to see the contacts in a sup. Figure. Its not clear how you can pack the "V1" domain crystals without interactions at the DF axle. Related to this the ADP

structure is generated by the addition of 2mM ADP, this is significantly higher than a natural concentration. How representative is this of the natural state. It would be good for the authors to state the calculated affinity for their "V1" domain to ATP and ADP to give the readers an insight into how much more is added for co-crystallisation. Also How did the authors check for bias within their structures given the molecular replacement approach on modest resolution data. I think they should state for clarity how they determined the level of bias, i.e. did they use poly alanine models and check for side chain density, use omitted parts of the structure to check the information was coming back etc... By using the same protein for molecular replacement it can be easy to obtain bias, although the larger movements may still be apparent more subtle changes maybe lost.

A: Thank you for your helpful and detailed suggestions. We have made a number of revisions (including new analyses and figures) to address these comments.

To clarify the crystal contacts, we have added a figure that shows the crystal packing contacts for soaked crystal structures (Supplementary Figure 5), and we have explained these results in the revised text (page 9, line 7; page 11, line 1 from bottom).

Additionally, we have described the binding affinities for nucleotides based on new data from an ITC analysis. Please also see our response to comment 1-1 of Referee #1.

Regarding the level of bias within the crystal structures, we cross-validated the results by obtaining various omit maps to minimize model bias. We have described this point in the Methods (page 25, line 3 from bottom) and main text (page 9, line 9; page 12, line 2), the omit maps of the shifted region are shown in Supplementary Figure 6.

Comment 2-3: It seems strange that the authors do not refer to the recent cryo-EM structures, especially that by John Rubinstein of the V-ATPase in three distinct states. How well do their structures compare? This would be a good way to back up the validity of a co-crystal structure on just part of a complex. It has also been shown in Rawson et al. that there is a significant amount of movement required to fit the central axle from the crystal structures into the cryo-EM map, reflective of some of the problems of crystallising an isolated domain rather than the whole complex. Of course the EM maps are not as high a resolution as shown here which is why comparing some of the EM data with the crystal structures could be a powerful approach.

A: Thank you for your suggestion. We have cited this reference⁴⁵ and compared the EM structures with our crystal structures in the revised text (page 18, line 8) and Supplementary Figure 10.

Comment 2-4: I can not find a legend for table 1.

A: Thank you for pointing out the lack of a legend for Table 1. We added this to the revised manuscript.

Comment 2-5: I see no need for or a top and bottom view in Figure 1. One would suffice and also cytoplasmic and luminal, or cytoplasmic and extracellular maybe more informative. Using N-terminal and C-terminals side in the figure legend is likely to be of little use to a wider audience.

A: We agree and have revised Figure 1 accordingly.

Comment 2-6: Both the abstract and introduction start by discussing the role of the V-ATPase in disease and its potential as a therapeutic target. However, this is not mentioned again in the manuscript and its unclear how this work is informing drug design. I would remove this.

A: We agree and have deleted these sentences in the revised manuscript.

Comment 2-7: Introduction, line 4, the authors compare the F and V-ATPase structures but fail to mention which subunits are from which complex. The sentence is also overlong.

A: This has been corrected in the revised manuscript.

Comment 2-8: r.m.s.d. values are given throughout the text but are these of the C α atoms or all atoms. Also given the resolution of $\sim 3\text{\AA}$, can r.m.s.d. values to 3 d.p. be justified?

A: Thank you for your suggestion. All r.m.s.d. values were calculated using C α atoms (we have described this on page 26, line 5) and we have reported all values to 2 decimal places.

Comment 2-9: Figure 3 legend the ADP is referred to as orange spheres, I would just state colored orange as there is they are not very spherical.

A: We agree and have revised the legend.

Comment 2-10: In Figure 4 the distances are shown but depending on the reproduction size of the figure I worry they will be unreadable.

A: We agree and have deleted the distances to avoid a misunderstanding. We have added the distances to the schematic binding sites in Supplementary Figure 7.

Comment 2-11: Figure 5 states "tight form (color) in ..." Which color?

A: We have revised the legend.

Comment 2-12: Also in the legend for figure 5 it states ADP is depicted as an orange stick, I think it would be better to refer to this as "in stick format, colored orange"

A: We agree and have revised the legend accordingly.

Comment 2-13: In the methods, crystallisation was carried out in "buffer b" what is buffer b? If this is a brand name from a certain supplier then please note this otherwise I am not sure how these experiments could be repeated.

A: We have added a description of this in the revised Methods.

Comment 2-14: I find the colour schemes at times confusing with the use of light and dark colours especially in the side views and superimpositions. For example Figure 2 a, c and e all the A and B domains look the same colour. The black and dark purple colours also provide little contrast. I would recommend more distinct colours to make it clearer to the reader.

A: Thank you for your advice. We have remade all figures using highly contrasting colours for clarity.

Comment 2-15: The references seem appropriate, however as discussed above I think a few references on the rotary ATPase family F/A & V would be beneficial (there are a number out there now with recent structures of all three).

A: We agree and have cited additional references on the rotary ATPase family in the revised manuscript.

Comment 2-16: In the main the clarity is good and the schematics of the proposed new catalytic cycle useful. It maybe difficult for people outside the direct field to follow but I am not sure if this is of much concern.

A: We agree and have revised the text that it is appropriate for a wide audience.

Reviewer #1 (Remarks to the Author):

The authors have included ITC data, which certainly strengthen the experimental basis. The AMPPNP data are straight-forward and convincing, but the ITC data for ADP binding are puzzling and appear over-interpreted as a three-site binding model on the basis of a single titration series (fig. 3b). These interactions must be elaborated by experimental variations that can qualify the interpretations of conformational changes associated with ADP binding.

Furthermore, to make a stronger point the soaking and ITC experiments should attempt combined ligand complexes (selecting from the relevant pool in various orders: ADP, AMPPNP, ATP, Pi, possibly also metallofluorides, nitrate, vanadate and what else - e.g. ADP added to the AMPPNP complex, AMPPNP or AlF₄⁻ added to ADP complex etc.). This may indeed be complex to interpret, but if the presented functional models and assumptions are correct, the observations and interpretations of mixed ligand complexes should also be possible and hold, and besides new insight it will add important internal controls of the functional models and single-ligand binding affinities. If not consistent, the functional model is not fulfilling and should not be published as presented, but limited to a more descriptive study.

Mixed ligands complexes may also be addressed by radioligand binding studies. Opportunities are again numerous, and the authors must devise an experimental strategy to support their claims. It is of course always possible to do more experiments, but in this case it seems critical to qualify interpretations of soaking experiments, which come with many possible pitfalls and biases that cannot be escaped by a single, successful ITC study.

So in conclusion I certainly miss a more solid ground of data to qualify a claim of an altered functional model of ATP hydrolysis in EhV1

Concerning unbiased observations and models of conformational changes I would further recommend isomorphous difference Fourier analysis, where conformational changes are also visualized on the basis of experimental difference coefficients (F_{obs}[1]-F_{obs}[2]) rather than model-derived coefficients only such as F_{obs}-F_{calc}. Pairs of isomorphous data sets for difference maps would at first approximation be selected by unit cell parameters (table 1), but optimally be scaling R-factor analysis.

Reviewer #2 (Remarks to the Author):

The paper by Murata and co workers shows a number of new crystal forms of the A1/V1 domain which shows new insights into its mechanism. This is novel research and provides new insights into the mechanical cycle, for example the release of phosphate before the ADP. The methodology is good but there are still some concerns on the crystal soaking which is renowned for not letting the full dynamics of a system to be seen. These limitations have now been referenced in the paper. The conclusions are based on these structures and in the main are valid although the problems with using high substrate concentrations and crystal soaking approaches remain. By showing the packing arrangements its now easier for the reader to assess themselves if crystal contacts may disrupt some of the conformational changes. The references are appropriate. The authors have addressed the critical reviews from before and have produced a more balanced manuscript. There are still some elements that could be better written such as "...induced conformational changes with changes in ..."

Title: Crystal structures of the ATP-binding and ADP-release dwells of the V₁ rotary motor

Authors: Kano Suzuki, Kenji Mizutani, Shintaro Maruyama, Kazumi Shimono, Fabiana L. Imai, Yoshimi Kakinuma, Yoshiko Ishizuka-Katsura, Mikako Shirouzu, Shigeyuki Yokoyama, Ichiro Yamato, and Takeshi Murata

Response to Referee Comments

Referee #1:

Comment 1-1: The authors have included ITC data, which certainly strengthen the experimental basis. The AMPPNP data are straight-forward and convincing, but the ITC data for ADP binding are puzzling and appear over-interpreted as a three-site binding model on the basis of a single titration series (fig. 3b). These interactions must be elaborated by experimental variations that can qualify the interpretations of conformational changes associated with ADP binding.

A: We agree that the ITC data for ADP binding are puzzling. We rephrased the sentences and added a supplementary explanation for the interpretation (page 13, line 4 from bottom).

Comment 1-2: Furthermore, to make a stronger point the soaking and ITC experiments should attempt combined ligand complexes (selecting from the relevant pool in various orders: ADP, AMPPNP, ATP, Pi, possibly also metallofluorides, nitrate, vanadate and what else - e.g. ADP added to the AMPPNP complex, AMPPNP or AlF₄⁻ added to ADP complex etc.). This may indeed be complex to interpret, but if the presented functional models and assumptions are correct, the observations and interpretations of mixed ligand

complexes should also be possible and hold, and besides new insight it will add important internal controls of the functional models and single-ligand binding affinities. If not consistent, the functional model is not fulfilling and should not be published as presented, but limited to a more descriptive study.

Mixed ligands complexes may also be addressed by radioligand binding studies. Opportunities are again numerous, and the authors must devise an experimental strategy to support their claims. It is of course always possible to do more experiments, but in this case it seems critical to qualify interpretations of soaking experiments, which come with many possible pitfalls and biases that cannot be escaped by a single, successful ITC study.

So in conclusion I certainly miss a more solid ground of data to qualify a claim of an altered functional model of ATP hydrolysis in EhV1

Concerning unbiased observations and models of conformational changes I would further recommend isomorphous difference Fourier analysis, where conformational changes are also visualized on the basis of experimental difference coefficients (Fobs[1]-Fobs[2]) rather than model-derived coefficients only such as Fobs-Fcalc. Pairs of isomorphous data sets for difference maps would at first approximation be selected by unit cell parameters (table 1), but optimally be scaling R-factor analysis.

A: Thank you for your helpful and detailed advice. We would like to perform additional crystal soaking, ITC, and radioligand binding experiments with various mixed ligands, not only to verify our mechanistic model, but also to elucidate the precise rotational mechanism of V₁-ATPase.

According to Referee#1's recommendation, we have performed an isomorphous difference Fourier analysis using Phenix (in isomorphous difference map mode) after re-indexing the cell dimensions in XDS for the calculations. Using the data for the ADP-bound structure (2_{ADP}V₁ or 3_{ADP}V₁), "Crystal symmetry mismatch" warnings were always obtained. To select pairs of the isomorphous data sets, the scores of CC (the linear correlation coefficient between normalized intensities E^2) between all pairs of data sets were calculated using POINTLESS in CCP4, and the results are following.

	eV_1	$2_{ATP}V_1$	$2_{ADP}V_1$	$3_{ADP}V_1$	$1_{Pi}V_1$
eV_1	1.000	0.666	0.216	0.064	0.658
$2_{ATP}V_1$	0.666	1.000	0.288	0.086	0.740
$2_{ADP}V_1$	0.216	0.288	1.000	0.053	0.343
$3_{ADP}V_1$	0.064	0.086	0.053	1.000	0.060
$1_{Pi}V_1$	0.658	0.740	0.343	0.060	1.000

Thus, the CC scores between data sets for ADP-bound Eh- V_1 ($2_{ADP}V_1$ or $3_{ADP}V_1$) and other Eh- V_1 are less than 0.35 because cell discrepancy between files is too large (i.e. they are not isomorphous). Therefore, we could not obtain reliable $F_{obs}-F_{obs}$ difference maps showing the conformational changes by ADP binding.

In order to verify the conformational changes by ADP binding, we have performed co-crystallisation of Eh- V_1 with ADP:Mg²⁺. However, we were not able to obtain the crystal structure of ADP-bound Eh- V_1 owing to the poor crystal qualities (low resolutions). This seems to be explained by the heterogeneity in purified Eh- V_1 by the addition of ADP; the binding affinity between the DF axis and A_3B_3 complex decreases and Eh- V_1 is not very stable, in the presence of ADP.

Recently, we successfully obtained the crystal structures of the AMP-PNP-bound and ADP-bound A_3B_3 mutants (cys A_3B_3 ; double cysteine mutant in the N-terminal β -barrel to stabilize the A_3B_3 complex by the disulfide bond), which were co-crystallised in the presence of 5 mM AMP-PNP and ADP, respectively (unpublished data). The structure of AMP-PNP-bound cys A_3B_3 showed two AMP-PNP molecules bound to two of three binding sites, and was almost identical to that of b A_3B_3 (wild-type A_3B_3 bound two AMP-PNP). In contrast, the structure of ADP-bound cys A_3B_3 showed that three ADP molecules were bound at all three binding sites similar as that of $3_{ADP}V_1$, as shown in the following figures.

Interestingly, the three AB forms of $3_{\text{ADP}} \text{cysA}_3\text{B}_3$ are slightly different from those of $3_{\text{ADP}} V_1$. Now we are analyzing these structural differences which should be generated by interaction with the DF complex. In any case, we think these findings are also supported by the conformational changes of Eh- V_1 induced by ADP binding. We hope that Referee #1 agrees with our model proposal by taking into account these unpublished data indicating the conformational changes of cysA_3B_3 by ADP binding, and the various omit maps to minimize model bias as shown in Supplementary Figure 6.

Referee #2:

Comment 2-1: The paper by Murata and co workers shows a number of new crystal forms of the A1/V1 domain which shows new insights into its mechanism. This is novel research and provides new insights into the mechanical cycle, for example the release of phosphate before the ADP. The methodology is good but there are still some concerns on the crystal soaking which is renowned for not letting the full dynamics of a system to be seen. These limitations have now been referenced in the paper. The conclusions are based on these structures and in the main are valid although the problems with using high substrate concentrations and crystal soaking approaches remain. By showing the packing arrangements its now easier for the reader to assess themselves if crystal contacts may disrupt some of the conformational changes. The references are appropriate. The authors have addressed the critical reviews from before ad have produced a more balanced manuscript.

A: Thank you for your kind comments.

Comment 2-2: There are still some elements that could be better written such as "...induced conformational changes with changes in the crystal packing interactions..."

A: We agree and have rephrased these sentences in the revised manuscript.

Title: Crystal structures of the ATP-binding and ADP-release dwells of the V₁ rotary motor

Authors: Kano Suzuki, Kenji Mizutani, Shintaro Maruyama, Kazumi Shimono, Fabiana L. Imai, Eiro Muneyuki, Yoshimi Kakinuma, Yoshiko Ishizuka-Katsura, Mikako Shirouzu, Shigeyuki Yokoyama, Ichiro Yamato, and Takeshi Murata

Response to Referee Comments

Referee #1:

Comment 1-1: The authors have included ITC data, which certainly strengthen the experimental basis. The AMPPNP data are straight-forward and convincing, but the ITC data for ADP binding are puzzling and appear over-interpreted as a three-site binding model on the basis of a single titration series (fig. 3b). These interactions must be elaborated by experimental variations that can qualify the interpretations of conformational changes associated with ADP binding.

A: We agree that the ITC data for ADP binding are puzzling. We have rephrased the relevant sentences and expanded on our explanation of the results (page 14, line 6 from the bottom). We have also performed additional ITC and tryptophan fluorescence experiments to evaluate these interpretations, as described in our answers to comments 1-2 and 1-3.

Comment 1-2: Furthermore, to make a stronger point the soaking and ITC experiments should attempt combined ligand complexes (selecting from the relevant pool in various orders: ADP, AMPPNP, ATP, Pi, possibly also metallofluorides, nitrate, vanadate and what else - e.g. ADP added to the AMPPNP complex, AMPPNP or AlF₄⁻ added to ADP complex etc.). This may indeed be complex to interpret, but if the presented functional models and assumptions are correct, the observations and interpretations of mixed ligand complexes should also be possible and hold, and besides new insight it will add important internal controls of the functional models and single-ligand binding affinities. If not

consistent, the functional model is not fulfilling and should not be published as presented, but limited to a more descriptive study.

A: Thank you for your helpful and detailed suggestions. We have rephrased part of our description of the mechanistic model of F- and V-ATPases to prevent over-interpretation in the revised manuscript. We have also made a number of revisions (including new analyses and figures) to address these comments as follows.

Regarding the soaking experiments:

We agree with this suggestion. We have attempted mixed-soaking experiments. However, we were unable to obtain the structures owing to the low resolution of the soaked crystals (worse than 6 Å), which probably reflects the heterogeneity of the bound substrates in the crystals. We have mentioned this in the revised text (page 13, line 7 from the bottom).

Regarding the ITC experiments:

We also performed displacement ITC experiments of ADP-bound and AMP-PNP-bound EhV₁ by addition of AMP-PNP and ADP, respectively. The findings are also consistent with the structural findings in this study. We have added them to the revised text (page 16, line 1).

Comment 1-3: Mixed ligands complexes may also be addressed by radioligand binding studies. Opportunities are again numerous, and the authors must devise an experimental strategy to support their claims. It is of course always possible to do more experiments, but in this case it seems critical to qualify interpretations of soaking experiments, which come with many possible pitfalls and biases that cannot be escaped by a single, successful ITC study. So in conclusion I certainly miss a more solid ground of data to qualify a claim of an altered functional model of ATP hydrolysis in EhV₁

A: We agree that it is necessary to qualify our interpretations of the soaking experiment results. In order to verify the conformational change induced by ADP binding observed by X-ray crystallography of ADP-soaked crystals, we measured the tryptophan

fluorescence of purified EhV₁ in the presence of AMP-PNP and/or ADP. Eh-A and Eh-B subunits of EhV₁ had 8 and 1 tryptophan residues, respectively, and showed high background fluorescence intensity. However, we detected reliable changes in fluorescence intensity after the addition of nucleotides. These findings are consistent with the ITC data and the structural results based on the soaking experiments. We have added them to the revised manuscript to clarify these points (page 17, lane 6):

Comment 1-4: Concerning unbiased observations and models of conformational changes I would further recommend isomorphous difference Fourier analysis, where conformational changes are also visualized on the basis of experimental difference coefficients (Fobs[1]-Fobs[2]) rather than model-derived coefficients only such as Fobs-Fcalc. Pairs of isomorphous data sets for difference maps would at first approximation be selected by unit cell parameters (table 1), but optimally be scaling R-factor analysis.

A: According to the recommendation of Referee #1, we have performed an isomorphous difference Fourier analysis using Phenix (in isomorphous difference map mode) after re-indexing the cell dimensions in XDS for the calculations. Using the data for the ADP-bound structure (2_{ADP}V₁ or 3_{ADP}V₁), “Crystal symmetry mismatch” warnings were always obtained. To select pairs of isomorphous data sets, the CC scores (i.e. the linear correlation coefficients between normalized intensities E^2) between all pairs of data sets were calculated using POINTLESS in CCP4, and the results are summarized in the table below.

	eV ₁	2 _{ATP} V ₁	2 _{ADP} V ₁	3 _{ADP} V ₁	1 _{Pi} V ₁
eV ₁	1.000	0.666	0.216	0.064	0.658
2 _{ATP} V ₁	0.666	1.000	0.288	0.086	0.740
2 _{ADP} V ₁	0.216	0.288	1.000	0.053	0.343
3 _{ADP} V ₁	0.064	0.086	0.053	1.000	0.060
1 _{Pi} V ₁	0.658	0.740	0.343	0.060	1.000

The CC scores for ADP-bound Eh-V₁ (2_{ADP}V₁ or 3_{ADP}V₁) and other Eh-V₁ were less than 0.35 because the cell discrepancy between files was too large (i.e. the data were not isomorphous). Therefore, we could not obtain reliable $F_{\text{obs}} - F_{\text{obs}}$ difference maps showing the conformational changes by ADP binding.

To verify the conformational changes by ADP binding, we also performed the co-crystallisation of Eh-V₁ with ADP:Mg²⁺. However, we were not able to obtain the crystal structure of ADP-bound Eh-V₁ owing to the poor crystal qualities (low resolutions). This may be explained by the heterogeneity in purified Eh-V₁ after the addition of ADP; the binding affinity between the DF axis and A₃B₃ complex decreases and Eh-V₁ is not very stable, in the presence of ADP. We hope that Referee #1 is satisfied with the revisions after considering our new ITC data and Trp fluorescence experiments as well as the various omit maps to minimize model bias, as shown in Supplementary Figure 5.

Referee #2:

Comment 2-1: The paper by Murata and co workers shows a number of new crystal forms of the A1/V1 domain which shows new insights into its mechanism. This is novel research and provides new insights into the mechanical cycle, for example the release of phosphate before the ADP. The methodology is good but there are still some concerns on the crystal soaking which is renowned for not letting the full dynamics of a system to be seen. These limitations have now been referenced in the paper. The conclusions are based on these structures and in the main are valid although the problems with using high substrate concentrations and crystal soaking approaches remain. By showing the packing arrangements its now easier for the reader to assess themselves if crystal contacts may disrupt some of the conformational changes. The references are appropriate. The authors have addressed the critical reviews from before ad have produced a more balanced manuscript.

A: Thank you for your kind comments.

Comment 2-2: There are still some elements that could be better written such as
"..induced conformational changes with changes in the crystal packing interactions..."

A: We agree and have carefully revised the manuscript for clarity.

Reviewer #1 (Remarks to the Author):

The authors have added valuable data to the soaking experiments, most notably further ITC analysis and Trp fluorescence data. The data points appear to be collected and interpreted correctly, and they add qualitative strength to the observations from soaking experiments.

However, I do not agree with the conclusion drawn on the failed experiments with mixed ligand soaking that may reflect more physiological settings of the ATPase. If these crystals diffract poorly because of mixed populations of binding modes it should in fact be a manageable problem to solve with a more careful attention to e.g. relative concentrations for pure binding at different sites, and/or time series of crystal flash cooling, similar to e.g.

<http://www.ncbi.nlm.nih.gov/pubmed/25664735> and
<http://www.ncbi.nlm.nih.gov/pmc/articles/PMC4839411/>

If poor diffraction properties on the other hand are due to conformational changes exceeding the flexibility of the crystal packing, then it tells us that important conformational changes take place that are not compatible with the experimental design, and the study loses the greater impact. A catch-22 for the authors unless physiologically relevant soaking experiment can be presented in this experimental design. If not, the study must limit itself to more descriptive conclusions.

Minor comments: many of the methodological contents are better fit into the data/results tables.

Reviewer #2 (Remarks to the Author):

The manuscript by Murata and co-workers is a detailed study of the catalytic cycle of the A1/V1 domain. There are important new insights here which would be important to both the V-ATPase and wider rotary ATPase field. The data are now well presented and after extensive revisions over two cycles of peer review I think the manuscript is now more robust and the conclusions sound based on the experimental approach. There are some limitations to the study, as there are with any crystal soaking experiments, but the authors have now made these clear and the results provide new insights. I don't see any need for further revisions/corrections.

First Response to Referees comments

Referee #1:

Comment 1-1: The data certainly appears to bear merit of new insight into the mechanism of V-ATPases, but the study seems too narrow and is based on the soaking studies alone, i.e. no complementary data to support the observations. Crystal soaking is a very useful approach when practically possible and validated, but one must also keep in mind that a crystal is a restrictive environment for conformational changes, which may not complete or perhaps be misleading, or not be observed at all due to crystal packing interactions overruling the ligand-induced changes. It would seem necessary to include other binding data than those observed in the soaked crystals, such as isothermal calorimetry, thermophoresis or surface plasmon resonance. Probing for conformational changes by e.g. Trp fluorescence may also be very helpful to qualify and validate the crystal observations.

A: Thank you for your advice. In response to this suggestion, we have performed isothermal titration calorimetry (ITC) experiments to estimate the binding properties of AMP-PNP and ADP, and have revised the text to include the ITC binding data (page 8, line 6; page 13, line 12; Figure 3; Supplementary Figure 3). We also performed a Trp fluorescence experiment to validate the crystal observations. However, we were not able to detect reliable fluorescence changes in response to the addition of nucleotides owing to high background Trp fluorescence attributed to 27 Trp residues in EhV₁ (Eh-A, 8 residues; Eh-B, 1 residue).

Comment 1-2: The manuscript is generally written in a fine command of English, but the flow of observations, interpretations, rationales and discussion that build up the dissemination are very difficult to follow for a non-expert in the field, which would be the the typical readership aimed for by this submission. A thorough revision is required, and should as mentioned also above include additional data to support the ligand-binding observations and induced changes.

A: We agree with your assessment of the manuscript structure and have revised the text extensively for readability. We have also added data to support the ligand-binding observations as mentioned in our response to comment 1-1.

Comment 1-3: The opening sentence of the abstract (V-ATPases as attractive drug target) is over-exposed and off-topic as this point is not addressed by the findings in the paper and does not serve to motivate the study otherwise. It should either be moderated, left out or addressed in the study.

A: We agree that the implications for drug target identification were not described in sufficient detail and did not serve to motivate the study. Accordingly, we have deleted the original opening sentence in the revised abstract.

Comment 1-4: "soaking [...] crystals into solution containing ..." (abstract and other places) sounds a bit odd and leaves the impression that crystals are dissolved.

A: We have corrected these phrases in the revised manuscript to avoid suggesting that the crystals were dissolved.

Comment 1-5: "inside of heaxogonally arranged [...] A3B3 and alpha3beta3...." (page 3) would be pseudo-hexagonally

A: Thank you for pointing out this error. It has been corrected in the revised manuscript.

Comment 1-6: Mixed soaks, e.g. ADP, Pi and AMPPNP altogether, would be very helpful to explore further the consistent sets of conformational changes that are relevant to the functional cycle.

A: We agree with this suggestion. We attempted mixed-soaking experiments. However, we were unable to obtain the structures owing to the low resolution of the soaked crystals (worse than 6 Å) so far, which probably reflects the heterogeneity of the bound substrates in the crystals.

Referee #2:

Comment 2-1: The first is the insistence on the authors calling this the V1 domain. There have been number of structural and biochemical studies which have shown a distinct difference between the F-ATPase, A-ATPase and V-ATPase. The structure studied here is of the A-ATPase family which are structurally and functionally distinct from the V-ATPase family. Indeed the paper makes no mention of the A-ATPase family. This structure being from the A-ATPase alone does not in my eyes diminish the importance, I think it still shows new novel insights, but there must be clarity in the field as to which system is which. If the authors believe strongly that this is a V not A-ATPase then this must be addressed fully within the paper for the readers to make up their own mind. They must also for clarity discuss the F/V and A-ATPase systems rather than just the F and V.

A: Thank you for raising this point regarding the differences between ATPase types. In the revised text, we have clarified the differences between F-, A-, and V-ATPases and justified referring to the structure as a homologue of eukaryotic V-ATPase (page 4, line 2; page 5, line 11).

Comment 2-2: Although there appears to be a distinct difference between this crystal form and others reported, this is still based on soaking and not co-crystallisation experiments. As such crystallisation artefacts maybe present. For example the authors talk about rotation of the DF axle, however is this compromised by the crystal contacts? It would be good to see the contacts in a sup. Figure. Its not clear how you can pack the "V1" domain crystals without interactions at the DF axle. Related to this the ADP structure is generated by the addition of 2mM ADP, this is significantly higher than a natural concentration. How representative is this of the natural state. It would be good for the authors to state the calculated affinity for their "V1" domain to ATP and ADP to give the readers an insight into how much more is added for co-crystallisation. Also How did the authors check for bias within their structures given the molecular replacement approach on modest resolution data. I think they should state for clarity

how they determined the level of bias, i.e. did they use poly alanine models and check for side chain density, use omitted parts of the structure to check the information was coming back etc... By using the same protein for molecular replacement it can be easy to obtain bias, although the larger movements may still be apparent more subtle changes maybe lost.

A: Thank you for your helpful and detailed suggestions. We have made a number of revisions (including new analyses and figures) to address these comments.

To clarify the crystal contacts, we have added a figure that shows the crystal packing contacts for soaked crystal structures (Supplementary Figure 5), and we have explained these results in the revised text (page 9, line 7; page 11, line 1 from bottom).

Additionally, we have described the binding affinities for nucleotides based on new data from an ITC analysis. Please also see our response to comment 1-1 of Referee #1.

Regarding the level of bias within the crystal structures, we cross-validated the results by obtaining various omit maps to minimize model bias. We have described this point in the Methods (page 25, line 3 from bottom) and main text (page 9, line 9; page 12, line 2), the omit maps of the shifted region are shown in Supplementary Figure 6.

Comment 2-3: It seems strange that the authors do not refer to the recent cryo-EM structures, especially that by John Rubinstein of the V-ATPase in three distinct states. How well do their structures compare? This would be a good way to back up the validity of a co-crystal structure on just part of a complex. It has also been shown in Rawson et al. that there is a significant amount of movement required to fit the central axle from the crystal structures into the cryo-EM map, reflective of some of the problems of crystallising an isolated domain rather than the whole complex. Of course the EM maps are not as high a resolution as shown here which is why comparing some of the EM data with the crystal structures could be a powerful approach.

A: Thank you for your suggestion. We have cited this reference⁴⁵ and compared the EM structures with our crystal structures in the revised text (page 18, line 8) and Supplementary Figure 10.

Comment 2-4: I can not find a legend for table 1.

A: Thank you for pointing out the lack of a legend for Table 1. We added this to the revised manuscript.

Comment 2-5: I see no need for or a top and bottom view in Figure 1. One would suffice and also cytoplasmic and luminal, or cytoplasmic and extracellular maybe more informative. Using N-terminal and C-terminals side in the figure legend is likely to be of little use to a wider audience.

A: We agree and have revised Figure 1 accordingly.

Comment 2-6: Both the abstract and introduction start by discussing the role of the V-ATPase in disease and its potential as a therapeutic target. However, this is not mentioned again in the manuscript and its unclear how this work is informing drug design. I would remove this.

A: We agree and have deleted these sentences in the revised manuscript.

Comment 2-7: Introduction, line 4, the authors compare the F and V-ATPase structures but fail to mention which subunits are from which complex. The sentence is also overlong.

A: This has been corrected in the revised manuscript.

Comment 2-8: r.m.s.d. values are given throughout the text but are these of the C α atoms or all atoms. Also given the resolution of $\sim 3\text{\AA}$, can r.m.s.d. values to 3 d.p. be justified?

A: Thank you for your suggestion. All r.m.s.d. values were calculated using C α atoms (we have described this on page 26, line 5) and we have reported all values to 2 decimal places.

Comment 2-9: Figure 3 legend the ADP is referred to as orange spheres, I would just state colored orange as there is they are not very spherical.

A: We agree and have revised the legend.

Comment 2-10: In Figure 4 the distances are shown but depending on the reproduction size of the figure I worry they will be unreadable.

A: We agree and have deleted the distances to avoid a misunderstanding. We have added the distances to the schematic binding sites in Supplementary Figure 7.

Comment 2-11: Figure 5 states "tight form (color) in ..." Which color?

A: We have revised the legend.

Comment 2-12: Also in the legend for figure 5 it states ADP is depicted as an orange stick, I think it would be better to refer to this as "in stick format, colored orange"

A: We agree and have revised the legend accordingly.

Comment 2-13: In the methods, crystallisation was carried out in "buffer b" what is buffer b? If this is a brand name from a certain supplier then please note this otherwise I am not sure how these experiments could be repeated.

A: We have added a description of this in the revised Methods.

Comment 2-14: I find the colour schemes at times confusing with the use of light and dark colours especially in the side views and superimpositions. For example Figure 2 a,

c and e all the A and B domains look the same colour. The black and dark purple colours also provide little contrast. I would recommend more distinct colours to make it clearer to the reader.

A: Thank you for your advice. We have remade all figures using highly contrasting colours for clarity.

Comment 2-15: The references seem appropriate, however as discussed above I think a few references on the rotary ATPase family F/A & V would be beneficial (there are a number out there now with recent structures of all three).

A: We agree and have cited additional references on the rotary ATPase family in the revised manuscript.

Comment 2-16: In the main the clarity is good and the schematics of the proposed new catalytic cycle useful. It maybe difficult for people outside the direct field to follow but I am not sure if this is of much concern.

A: We agree and have revised the text that it is appropriate for a wide audience.

Second Response to Referee Comments

Referee #1:

Comment 1-1: The authors have included ITC data, which certainly strengthen the experimental basis. The AMPPNP data are straight-forward and convincing, but the ITC data for ADP binding are puzzling and appear over-interpreted as a three-site binding model on the basis of a single titration series (fig. 3b). These interactions must be elaborated by experimental variations that can qualify the interpretations of conformational changes associated with ADP binding.

A: We agree that the ITC data for ADP binding are puzzling. We have rephrased the relevant sentences and expanded on our explanation of the results (page 14, line 6 from the bottom). We have also performed additional ITC and tryptophan fluorescence experiments to evaluate these interpretations, as described in our answers to comments 1-2 and 1-3.

Comment 1-2: Furthermore, to make a stronger point the soaking and ITC experiments should attempt combined ligand complexes (selecting from the relevant pool in various orders: ADP, AMPPNP, ATP, Pi, possibly also metallofluorides, nitrate, vanadate and what else - e.g. ADP added to the AMPPNP complex, AMPPNP or AlF₄⁻ added to ADP complex etc.). This may indeed be complex to interpret, but if the presented functional models and assumptions are correct, the observations and interpretations of mixed ligand complexes should also be possible and hold, and besides new insight it will add important internal controls of the functional models and single-ligand binding affinities. If not consistent, the functional model is not fulfilling and should not be published as presented, but limited to a more descriptive study.

A: Thank you for your helpful and detailed suggestions. We have rephrased part of our description of the mechanistic model of F- and V-ATPases to prevent over-interpretation in the revised manuscript. We have also made a number of revisions (including new analyses and figures) to address these comments as follows.

Regarding the soaking experiments:

We agree with this suggestion. We have attempted mixed-soaking experiments. However, we were unable to obtain the structures owing to the low resolution of the soaked crystals (worse than 6 Å), which probably reflects the heterogeneity of the bound substrates in the crystals. We have mentioned this in the revised text (page 13, line 7 from the bottom).

Regarding the ITC experiments:

We also performed displacement ITC experiments of ADP-bound and AMP-PNP-bound EhV₁ by addition of AMP-PNP and ADP, respectively. The findings are also consistent with the structural findings in this study. We have added them to the revised text (page 16, line 1).

Comment 1-3: Mixed ligands complexes may also be addressed by radioligand binding studies. Opportunities are again numerous, and the authors must devise an experimental strategy to support their claims. It is of course always possible to do more experiments, but in this case it seems critical to qualify interpretations of soaking experiments, which come with many possible pitfalls and biases that cannot be escaped by a single, successful ITC study. So in conclusion I certainly miss a more solid ground of data to qualify a claim of an altered functional model of ATP hydrolysis in EhV1

A: We agree that it is necessary to qualify our interpretations of the soaking experiment results. In order to verify the conformational change induced by ADP binding observed by X-ray crystallography of ADP-soaked crystals, we measured the tryptophan fluorescence of purified EhV₁ in the presence of AMP-PNP and/or ADP. Eh-A and Eh-B subunits of EhV₁ had 8 and 1 tryptophan residues, respectively, and showed high background fluorescence intensity. However, we detected reliable changes in fluorescence intensity after the addition of nucleotides. These findings are consistent with the ITC data and the structural results based on the soaking experiments. We have added them to the revised manuscript to clarify these points (page 17, lane 6):

Comment 1-4: Concerning unbiased observations and models of conformational changes I would further recommend isomorphous difference Fourier analysis, where conformational changes are also visualized on the basis of experimental difference coefficients ($F_{obs}[1]-F_{obs}[2]$) rather than model-derived coefficients only such as $F_{obs}-F_{calc}$. Pairs of isomorphous data sets for difference maps would at first approximation be selected by unit cell parameters (table 1), but optimally be scaling R-factor analysis.

A: According to the recommendation of Referee #1, we have performed an isomorphous difference Fourier analysis using Phenix (in isomorphous difference map mode) after re-indexing the cell dimensions in XDS for the calculations. Using the data for the ADP-bound structure ($2_{ADP}V_1$ or $3_{ADP}V_1$), “Crystal symmetry mismatch” warnings were always obtained. To select pairs of isomorphous data sets, the CC scores (i.e. the linear correlation coefficients between normalized intensities E^2) between all pairs of data sets were calculated using POINTLESS in CCP4, and the results are summarized in the table below.

	eV_1	$2_{ATP}V_1$	$2_{ADP}V_1$	$3_{ADP}V_1$	$1_{Pi}V_1$
eV_1	1.000	0.666	0.216	0.064	0.658
$2_{ATP}V_1$	0.666	1.000	0.288	0.086	0.740
$2_{ADP}V_1$	0.216	0.288	1.000	0.053	0.343
$3_{ADP}V_1$	0.064	0.086	0.053	1.000	0.060
$1_{Pi}V_1$	0.658	0.740	0.343	0.060	1.000

The CC scores for ADP-bound Eh- V_1 ($2_{ADP}V_1$ or $3_{ADP}V_1$) and other Eh- V_1 were less than 0.35 because the cell discrepancy between files was too large (i.e. the data were not isomorphous). Therefore, we could not obtain reliable $F_{obs}-F_{obs}$ difference maps showing the conformational changes by ADP binding.

To verify the conformational changes by ADP binding, we also performed the co-crystallisation of Eh- V_1 with $ADP:Mg^{2+}$. However, we were not able to obtain the

crystal structure of ADP-bound Eh-V₁ owing to the poor crystal qualities (low resolutions). This may be explained by the heterogeneity in purified Eh-V₁ after the addition of ADP; the binding affinity between the DF axis and A₃B₃ complex decreases and Eh-V₁ is not very stable, in the presence of ADP. We hope that Referee #1 is satisfied with the revisions after considering our new ITC data and Trp fluorescence experiments as well as the various omit maps to minimize model bias, as shown in Supplementary Figure 5.

Referee #2:

Comment 2-1: The paper by Murata and co workers shows a number of new crystal forms of the A1/V1 domain which shows new insights into its mechanism. This is novel research and provides new insights into the mechanical cycle, for example the release of phosphate before the ADP. The methodology is good but there are still some concerns on the crystal soaking which is renowned for not letting the full dynamics of a system to be seen. These limitations have now been referenced in the paper. The conclusions are based on these structures and in the main are valid although the problems with using high substrate concentrations and crystal soaking approaches remain. By showing the packing arrangements its now easier for the reader to assess themselves if crystal contacts may disrupt some of the conformational changes. The references are appropriate. The authors have addressed the critical reviews from before ad have produced a more balanced manuscript.

A: Thank you for your kind comments.

Comment 2-2: There are still some elements that could be better written such as "...induced conformational changes with changes in the crystal packing interactions..."

A: We agree and have carefully revised the manuscript for clarity.

Final RESPONSE TO REFEREE COMMENTS

Referee #1:

Comment 1-1: The authors have added valuable data to the soaking experiments, most notably further ITC analysis and Trp fluorescence data. The data points appear to be collected and interpreted correctly, and they add qualitative strength to the observations from soaking experiments.

However, I do not agree with the conclusion drawn on the failed experiments with mixed ligand soaking that may reflect more physiological settings of the ATPase. If these crystals diffract poorly because of mixed populations of binding modes it should in fact be a manageable problem to solve with a more careful attention to e.g. relative concentrations for pure binding at different sites, and/or time series of crystal flash cooling, similar to e.g. <http://www.ncbi.nlm.nih.gov/pubmed/25664735> and <http://www.ncbi.nlm.nih.gov/pmc/articles/PMC4839411/>

If poor diffraction properties on the other hand are due to conformational changes exceeding the flexibility of the crystal packing, then it tells us that important conformational changes take place that are not compatible with the experimental design, and the study loses the greater impact. A catch-22 for the authors unless physiologically relevant soaking experiment can be presented in this experimental design. If not, the study must limit itself to more descriptive conclusions.

A: Thank you for your helpful and detailed advice. We believe the poor diffraction can be attributed to the former explanation. We would like to optimize the ligand concentrations and crystal soaking times until flash cooling in the near future, not only to verify our mechanistic model, but also to elucidate the precise rotational mechanism of V_1 -ATPase. We have rephrased the text related to the failed mixed ligand experiments to clarify these points in the revised manuscript as follows (page 13, line 5 from the bottom).

We also soaked the crystals of eV_1 in the mixture of various concentrations of AMP-PNP, ADP, and/or Pi to obtain other intermediate states. However, diffraction of these soaked crystals was not sufficient to solve the structure. Careful optimization of the ligand concentrations and crystal soaking times are necessary to improve the resolutions.

Comment 1-2: many of the methodological contents are better fit into the data/results tables.

A: According to your comment, the relevant methodological contents have been moved or added to Table 1 and the legends of Figure 8 and Supplementary Table 1.

Referee #2:

Comment 2-1: The manuscript by Murata and co-workers is a detailed study of the catalytic cycle of the A1/V1 domain. There are important new insights here which would be important to both the V-ATPase and wider rotary ATPase field. The data are now well presented and after extensive revisions over two cycles of peer review I think the manuscript is now more robust and the conclusions sound based on the experimental approach. There are some limitations to the study, as there are with any crystal soaking experiments, but the authors have now made these clear and the results provide new insights. I dont see any need for further revisons/corrections.

A: Thank you for your kind comments.